# Neural assemblies uncovered by generative modeling explain whole-brain activity statistics and reflect structural connectivity

Thijs L van der Plas[1,2,3†], Jérôme Tubiana[4†], Guillaume Le Goc[2], Geoffrey Migault[2], Michael Kunst[5,6], Herwig Baier[5], Volker Bormuth[2‡], Bernhard Englitz[1‡], Georges Debrégeas[2*‡]

[1]Computational Neuroscience Lab, Department of Neurophysiology, Donders Center for Neuroscience, Radboud University, Nijmegen, Netherlands; [2]Sorbonne Université, CNRS, Institut de Biologie Paris-Seine (IBPS), Laboratoire Jean Perrin (LJP), Paris, France; [3]Department of Physiology, Anatomy and Genetics, University of Oxford, Oxford, United Kingdom; [4]Blavatnik School of Computer Science, Tel Aviv University, Tel Aviv, Israel; [5]Department Genes – Circuits – Behavior, Max Planck Institute for Biological Intelligence, Martinsried, Germany; [6]Allen Institute for Brain Science, Seattle, United States

*For correspondence: georges.debregeas@sorbonne-universite.fr

†These authors contributed equally to this work
‡These authors also contributed equally to this work

**Competing interest:** The authors declare that no competing interests exist.

**Abstract** Patterns of endogenous activity in the brain reflect a stochastic exploration of the neuronal state space that is constrained by the underlying assembly organization of neurons. Yet, it remains to be shown that this interplay between neurons and their assembly dynamics indeed suffices to generate whole-brain data statistics. Here, we recorded the activity from ~40,000 neurons simultaneously in zebrafish larvae, and show that a data-driven generative model of neuron-assembly interactions can accurately reproduce the mean activity and pairwise correlation statistics of their spontaneous activity. This model, the compositional Restricted Boltzmann Machine (cRBM), unveils ~200 neural assemblies, which compose neurophysiological circuits and whose various combinations form successive brain states. We then performed in silico perturbation experiments to determine the interregional functional connectivity, which is conserved across individual animals and correlates well with structural connectivity. Our results showcase how cRBMs can capture the coarse-grained organization of the zebrafish brain. Notably, this generative model can readily be deployed to parse neural data obtained by other large-scale recording techniques.

## Editor's evaluation

Large scale recordings, sometimes involving 10s of thousands of neurons, are becoming increasingly common. Making sense of these recordings, however, is not easy. This paper introduces a new method, the compositional Restricted Boltzmann Machine, that overcomes this problem -- it can find structure in data, including both "cell assemblies" and structural connectivity, without inordinate computing resources (data from 40,000 neurons recorded from zebrafish can be analyzed in less than a day). This is a valuable contribution, both to those interested in data analysis, and to those interested in zebrafish.

## Introduction

The brain is a highly connected network, organized across multiple scales, from local circuits involving just a few neurons to extended networks spanning multiple brain regions (*White et al., 1986*; *Song et al., 2005*; *Kunst et al., 2019*). Concurrent with this spatial organization, brain activity exhibits correlated firing among large groups of neurons, often referred to as neural assemblies (*Harris, 2005*). This assembly organization of brain dynamics has been observed in, for example, auditory cortex (*Bathellier et al., 2012*), motor cortex (*Narayanan et al., 2005*), prefrontal cortex (*Tavoni et al., 2017*), hippocampus (*Lin et al., 2005*), retina (*Shlens et al., 2009*), and zebrafish optic tectum (*Romano et al., 2015*; *Mölter et al., 2018*; *Diana et al., 2019*; *Triplett et al., 2020*). These neural assemblies are thought to form elementary computational units and subserve essential cognitive functions such as short-term memory, sensorimotor computation or decision-making (*Hebb, 1949*; *Gerstein et al., 1989*; *Harris, 2005*; *Buzsáki, 2010*; *Harris, 2012*; *Palm et al., 2014*; *Eichenbaum, 2018*). Despite the prevalence of these assemblies across the nervous system and their role in neural computation, it remains an open challenge to extract the assembly organization of a full brain and to show that the assembly activity state, derived from that of the neurons, is sufficient to account for the collective neural dynamics.

The need to address this challenge is catalyzed by technological advances in light-sheet micros-copy, enabling the simultaneous recording of the majority of neurons in the zebrafish brain at single-cell resolution in vivo (*Panier et al., 2013*; *Ahrens et al., 2013*; *Wolf et al., 2015*; *Wolf et al., 2017*; *Migault et al., 2018*; *Vanwalleghem et al., 2018*). This neural recording technique opens up new avenues for constructing near-complete models of neural activity, and in particular its assembly organization. Recent attempts have been made to identify assemblies using either clustering (*Panier et al., 2013*; *Triplett et al., 2018*; *Chen et al., 2018*; *Mölter et al., 2018*; *Bartoszek et al., 2021*), dimensionality reduction approaches (*Lopes-dos-Santos et al., 2013*; *Romano et al., 2015*; *Mu et al., 2019*) or latent variable models (*Diana et al., 2019*; *Triplett et al., 2020*), albeit often limited to single brain regions. However, these methods do not explicitly assess to what extent the inferred assemblies could give rise to the observed neural data statistics, which is a crucial property of physiologically meaningful assemblies (*Harris, 2005*). Here, we address this challenge by developing a generative model of neural activity that is explicitly constrained by the assembly organization, thereby quantifying if assemblies indeed suffice to produce the observed neural data statistics.

Specifically, we formalize neural assemblies using a bipartite network of two connected layers representing the neuronal and the assembly activity, respectively. Together with the maximum entropy principle (*Jaynes, 1957*; *Bialek, 2012*), this architecture defines the Restrictive Boltzmann Machine (RBM) model (*Hinton and Salakhutdinov, 2006*). Here, we use an extension to the classical RBM definition termed compositional RBM (cRBM) that we have recently introduced (*Tubiana and Monasson, 2017*; *Tubiana et al., 2019a*) and which brings multiple advances to assembly-based network modeling: (1) The maximum entropy principle ensures that neural assemblies are inferred solely from the data statistics. (2) The generative nature of the model, through alternate data sampling of the neuronal and assembly layers, can be leveraged to evaluate its capacity to replicate the empirical data statistics, such as the pairwise co-activation probabilities of all neuron pairs. (3) The cRBM steers the assembly organization to the so-called compositional phase where a small number of assemblies are active at any point in time, making the resulting model highly interpretable as we have shown previously for protein sequence analysis (*Tubiana et al., 2019b*).

Here, we have successfully trained cRBMs to brain-scale, neuron-level recordings of spontaneous activity in larval zebrafish containing 41,000 neurons on average (*Panier et al., 2013*; *Wolf et al., 2017*; *Migault et al., 2018*). This represents an increase of ~2 orders of magnitude in number of neurons with respect to previously reported RBM implementations (*Köster et al., 2014*; *Gardella et al al., 2017*; *Volpi et al., 2020*), attained through significant algorithmic and computational enhancements. We found that all cells could be grouped into 100–200 partially overlapping assemblies, which are anatomically localized and together span the entire brain, and accurately replicate the first and second order statistics of the neural activity. These assemblies were found to carry more predictive power than a fully connected model which has orders of magnitude more parameters, validating that assemblies underpin collective neural dynamics. Further, the probabilistic nature of our model allowed us to compute a functional connectivity matrix by quantifying the effect of activity

perturbations in silico. This assembly-based functional connectivity is well-conserved across individual fish and consistent with anatomical connectivity at the mesoscale (*Kunst et al., 2019*).

In summary, we present an assembly decomposition spanning the zebrafish brain, which accurately accounts for its activity statistics. Our cRBM model provides a widely applicable tool to the community to construct low-dimensional data representations that are defined by the statistics of the data, in particular for very high-dimensional systems. Its generative capability further allows to produce new (synthetic) activity patterns that are amenable to direct in silico perturbation and ablation studies.

## Results

### Compositional RBMs construct Hidden Units by grouping neurons into assemblies

Spontaneous neural activity was recorded from eight zebrafish larvae aged 5–7 days post fertilization expressing the GCaMP6s or GCaMP6f calcium reporters using light-sheet microscopy (*Panier et al., 2013*; *Wolf et al., 2017*; *Migault et al., 2018*). Each data set contained the activity of a large fraction of the neurons in the brain ($40709 \pm 13854$; mean ± standard deviation), which, after cell segmentation, were registered onto the ZBrain atlas (*Randlett et al., 2015*) and mapzebrain atlas (*Kunst et al., 2019*). Individual neuronal fluorescence traces were deconvolved to binarized spike trains using blind sparse deconvolution (*Tubiana et al., 2020*). This data acquisition process is depicted in *Figure 1A*.

We trained compositional Restricted Boltzmann Machine (cRBM) models to capture the activity statistics of these neural recordings. cRBMs are maximum entropy models, that is, the maximally unconstrained solution that fits model-specific data statistics (*Hinton and Salakhutdinov, 2006*; *Tubiana and Monasson, 2017*; *Gardella et al., 2019*), and critically extend the classical RBM formulation. Its architecture consists of a bipartite graph where the high-dimensional layer of neurons $\mathbf{v}$ (named 'visible units' in RBM terminology) is connected to the low-dimensional layer of latent components, termed Hidden Units (HUs) $\mathbf{h}$. Their interaction is characterized by a weight matrix $\mathbf{W}$ that is regularized to be sparse. The collection of neurons that have non-zero interactions with a particular HU, noted $h_\mu$ (i.e. with $|w_{i,\mu}| > 0$), define its corresponding neural assembly μ (*Figure 1B*). This weight matrix, together with the neuron weight vector $\mathbf{g}$ and HU potential $\mathcal{U}$, defines the transformation from the binarized neural activity $\mathbf{v}(t)$ to the continuous HU activity $\mathbf{h}(t)$ (*Figure 1B*). *Figure 1C* shows all recorded neurons of a zebrafish brain, color-labeled according to their strongest-connecting HU, illustrating that cRBM-inferred assemblies (hereafter, neural assemblies for conciseness) are typically densely localized in space and together span the entire brain.

Beyond its architecture (*Figure 2A*), the model is defined by the probability function $P(\mathbf{v}, \mathbf{h})$ of any data configuration $(\mathbf{v}, \mathbf{h})$ (see Materials and methods 'Restricted Boltzmann Machines' and 'Compositional Restricted Boltzmann Machine' for details):

$$P(\mathbf{v}, \mathbf{h}) = \frac{1}{Z} \exp\left(-E(\mathbf{v}, \mathbf{h})\right) \tag{1}$$

where $Z$ is the partition function that normalizes *Equation 1* and $E$ is the following energy function:

$$E(\mathbf{v}, \mathbf{h}) = -\sum_i g_i v_i + \sum_\mu \mathcal{U}_\mu(h_\mu) - \sum_{i,\mu} w_{i,\mu} v_i h_\mu \tag{2}$$

HU activity h is obtained by sampling from the conditional probability function $P(\mathbf{h}|\mathbf{v})$:

$$P(\mathbf{h}|\mathbf{v}) = \prod_{\mu=1}^{M} P(h_\mu|\mathbf{v}) \propto \prod_{\mu=1}^{M} \exp\left(-\mathcal{U}_\mu(h_\mu) + h_\mu \cdot \sum_i w_{i,\mu} v_i\right) \tag{3}$$

Conversely, neural activity is obtained from HU activity through:

$$P(\mathbf{v}|\mathbf{h}) = \prod_{i=1}^{N} P(v_i|\mathbf{h}) \propto \prod_{i=1}^{N} \exp\left(g_i v_i + v_i \cdot \sum_\mu w_{i,\mu} h_\mu\right) \tag{4}$$

*Equations 3 and 4* mathematically reflect the dual relationship between neural and assembly states: the Hidden Units $\mathbf{h}$ drive 'visible' neural activity $\mathbf{v}$, expressed as $P(\mathbf{v}|\mathbf{h})$, while the stochastic assembly

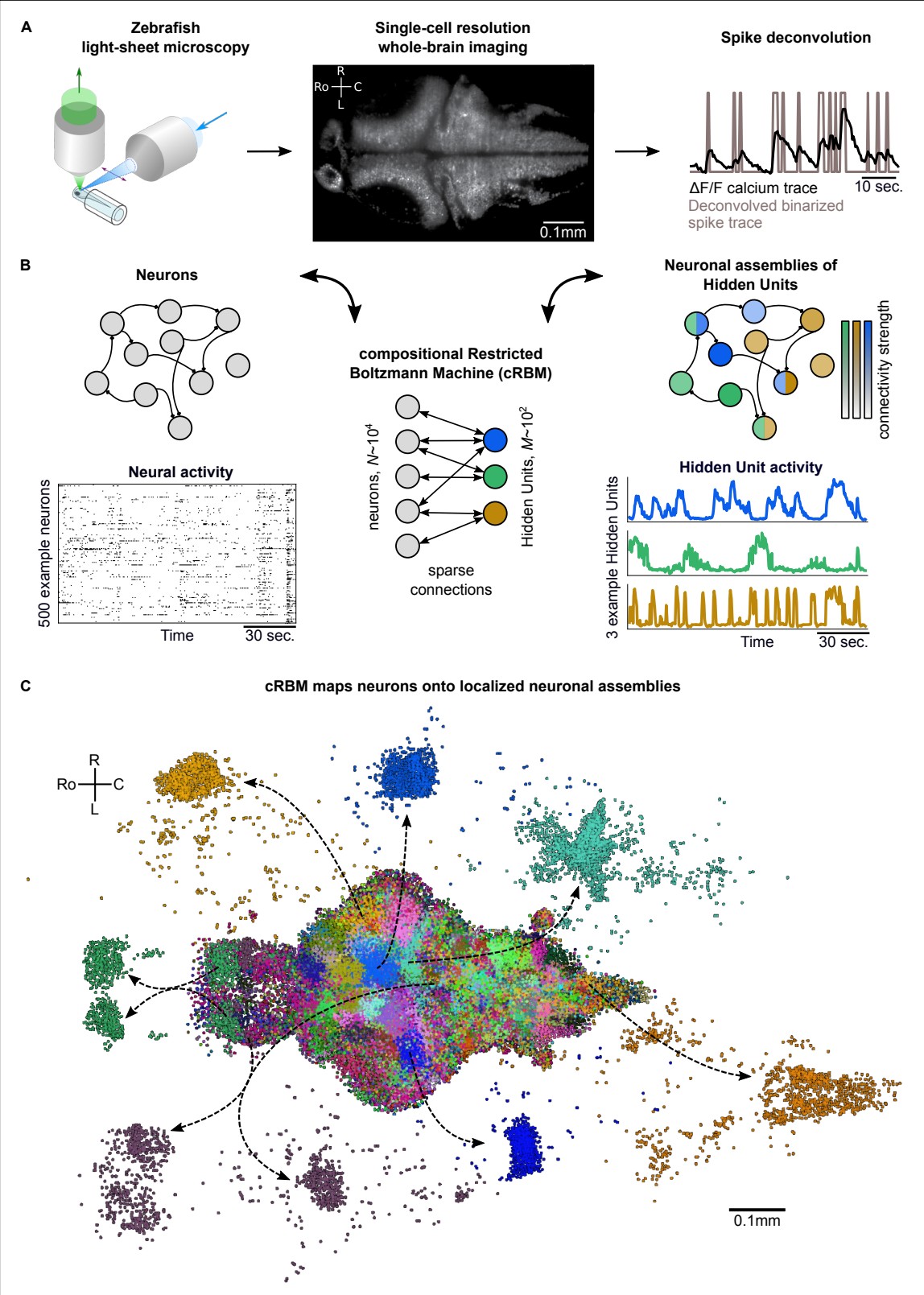

**Figure 1.** cRBMs construct Hidden Units by grouping neurons into assemblies. (**A**) The neural activity of zebrafish larvae was imaged using light-sheet microscopy (left), which resulted in brain-scale, single-cell resolution data sets (middle, microscopy image of a single plane shown for fish #1). Calcium activity $\Delta F/F$ was deconvolved to binarized spike traces for each segmented cell (right, example neuron). (**B**) cRBM sparsely connects neurons (left) to Hidden Units (HUs, right). The neurons that connect to a given HU (and thus belong to the associated assembly), are depicted by the corresponding

*Figure 1 continued on next page*

*Figure 1 continued*

color labeling (right panel). Data sets typically consist of $N \sim 10^4$ neurons and $M \sim 10^2$ HUs. The activity of 500 randomly chosen example neurons (raster plot, left) and HUs 99, 26, 115 (activity traces, right) of the same time excerpt is shown. HU activity is continuous and is determined by transforming the neural activity of its assembly. (**C**) The neural assemblies of an example data set (fish #3) are shown by coloring each neuron according to its strongest-connecting HU. 7 assemblies are highlighted (starting rostrally at the green forebrain assembly, going clockwise: HU 177, 187, 7, 156, 124, 64, 178), by showing their neurons with a connection $|w| \geq 0.1$. See *Figure 3* for more anatomical details of assemblies. R: Right, L: Left, Ro: Rostral, C: Caudal.

activity **h** itself is defined as a function of the activity of the neurons: $P(\mathbf{h}|\mathbf{v})$. Importantly, the model does not include direct connections between neurons, hence neural correlations $\langle v_i v_j \rangle$ can arise solely from shared assemblies. Moreover, this bipartite architecture ensures that the conditional distributions factorize, leading to a sampling procedure where all neurons or all HUs can be sampled in parallel. The cRBM leverages this property to efficiently generate new data by Monte Carlo sampling alternately from $P(\mathbf{h}|\mathbf{v})$ and $P(\mathbf{v}|\mathbf{h})$ (*Figure 2B*).

The cRBM differs from the classical RBM formulation (*Hinton and Salakhutdinov, 2006*) through the introduction of double Rectified Linear Unit (dReLU) potentials $\mathcal{U}_\mu$, weight sparsity regularization and normalized HU activity (further detailed in Methods). We have previously demonstrated in theory and application (*Tubiana and Monasson, 2017*; *Tubiana et al., 2019a*; *Tubiana et al., 2019b*) that this new formulation steers the model into the so-called compositional phase, which makes the latent representation highly interpretable. This phase occurs when a limited number $m$ of HUs co-activate such that $1 \ll m \ll M$ where $M$ is the total number of HUs. Thus, each visible configuration is mapped to a specific combination of activated HUs. This contrasts with the ferromagnetic phase ($m \sim 1$) where each HU encodes one specific activity pattern, thus severely limiting the possible number of encoded patterns, or the spin-glass phase ($m \sim M$) where all HUs activate simultaneously, yielding a very complex assembly patchwork (*Tubiana and Monasson, 2017*). Therefore, the compositional phase can provide the right level of granularity for a meaningful interpretation of the cRBM neural assemblies by decomposing the overall activity as a time-dependent co-activation of different assemblies of interpretable size and extent.

## Trained cRBMs accurately replicate data statistics

cRBM models are trained to maximize the $P(\mathbf{v}, \mathbf{h})$ log-likelihood of the zebrafish data recordings, which is achieved by matching the model-generated statistics $\langle v_i \rangle$, $\langle h_\mu \rangle$ and $\langle v_i h_\mu \rangle$ (the mean neuronal activity, mean HU activity and their correlations, respectively) to the empirical data statistics (*Equation 14*). In order to optimize the two free parameters of the cRBM model – the sparsity regularization parameter $\lambda$ and the total number of HUs $M$ – we assessed the cRBM performance for a grid of $(\lambda, M)$-values for one data set (fish #3). This analysis yielded an optimum for $\lambda = 0.02$ and $M = 200$ (*Figure 2—figure supplement 1*). These values were subsequently used for all recordings, where $M$ was scaled with the number of neurons $N$.

We trained cRBMs on 70% of the recording length, and compared the statistics of model-generated data to the withheld test data set (the remaining 30% of recording, see Materials and methods 'Train / test data split' and 'Assessment of data statistics' for details). After convergence, the cRBM generated data that replicated the training statistics accurately, with normalized Root Mean Square Error (nRMSE) values of $\text{nRMSE}_{\langle v_i \rangle} = 0.11$, $\text{nRMSE}_{\langle h_\mu \rangle} = 0.15$ and $\text{nRMSE}_{\langle v_i h_\mu \rangle} = 0.09$ (*Figure 2C-E*). Here, nRMSE is normalized such that 1 corresponds to shuffled data statistics and 0 corresponds to the best possible RMSE, i.e., between train and test data.

We further evaluated cRBM performance to assess its ability to capture data statistics that the cRBM was not explicitly trained to replicate: the pairwise correlations between neurons $\langle v_i v_j \rangle$ and the pairwise correlations between HUs $\langle h_\mu h_\nu \rangle$. We found that these statistics were also accurately replicated by model-generated data, with $\text{nRMSE}_{\langle v_i v_j \rangle} = -0.09$ (meaning that the model slightly outperformed the train-test data difference) and $\text{nRMSE}_{\langle h_\mu h_\nu \rangle} = 0.17$ (*Figure 2F, G*). The fact that cRBM also accurately replicated neural correlations $\langle v_i v_j \rangle$ (*Figure 2F*) is of particular relevance, since this indicates that (1) the assumption that neural correlations can be explained by their shared assemblies is justified and (2) cRBMs may provide an efficient mean to model neural interactions of such large systems ($N \sim 10^4$) where directly modeling all $N^2$ interactions would be computationally infeasible or not sufficiently constrained by the available data.

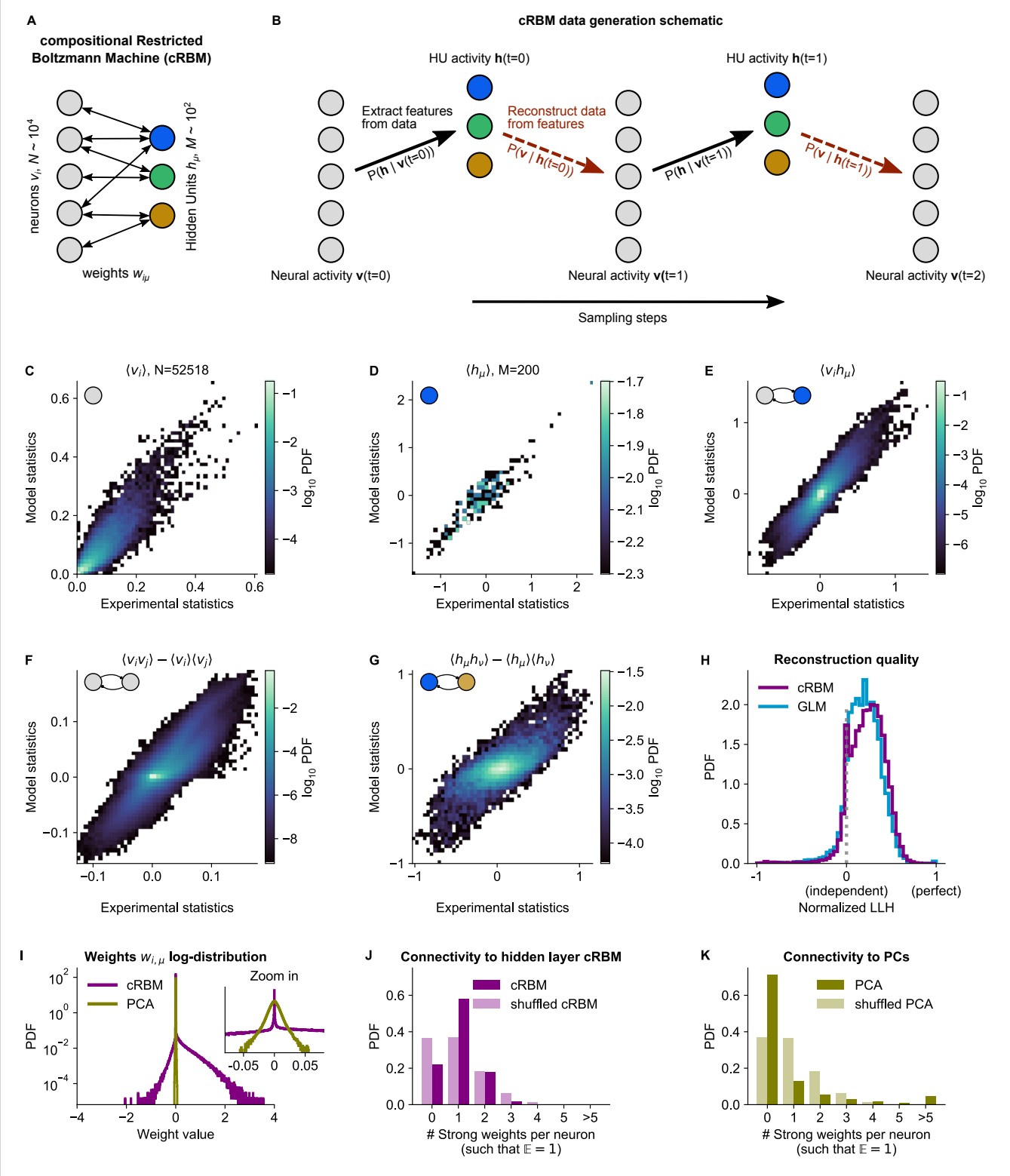

**Figure 2.** cRBM is optimized to accurately replicate data statistics. (**A**) Schematic of the cRBM architecture, with neurons $v_i$ on the left, HUs $h_\mu$ on the right, connected by weights $w_{i,\mu}$. (**B**) Schematic depicting how cRBMs generate new data. The HU activity $\mathbf{h}(t)$ is sampled from the visible unit (i.e. neuron) configuration $\mathbf{v}(t)$, after which the new visible unit configuration $\mathbf{v}(t+1)$ is sampled and so forth. (**C**) cRBM-predicted and experimental mean neural activity $\langle v_i \rangle$ were highly correlated (Pearson correlation $r_\mathbf{P} = 0.91$, $P < 10^{-307}$) and had low error (nRMSE$_{\langle v_i \rangle} = 0.11$, normalized Root Mean Square Error, see Materials and methods - 'Calculating the normalized Root Mean Square Error' ). Data displayed as 2D probability density function

*Figure 2 continued*

(PDF), scaled logarithmically (base 10). (**D**) cRBM-predicted and experimental mean Hidden Unit (HU) activity also correlated very strongly ($r_P = 0.93$, $P < 10^{-86}$) and had low $\mathrm{nRMSE}_{\langle h_\mu \rangle} = 0.15$ (other details as in C) (**E**) cRBM-predicted and experimental average pairwise neuron-HU interactions $\langle v_i h_\mu \rangle$ correlated strongly ($r_P = 0.74$, $P < 10^{-307}$) and had a low error ($\mathrm{nRMSE}_{\langle v_i h_\mu \rangle} = 0.09$). (**F**) cRBM-predicted and experimental average pairwise neuron-neuron interactions $\langle v_i v_j \rangle$ correlated well ($r_P = 0.58$, $P < 10^{-307}$) and had a low error ($\mathrm{nRMSE}_{\langle v_i v_j \rangle} = -0.09$, where the negative nRMSE value means that cRBM-predictions match the test data slightly better than the train data). Pairwise interactions were corrected for naive correlations due to their mean activity by subtracting $\langle v_i \rangle \langle v_j \rangle$. (**G**) cRBM-predicted and experimental average pairwise HU-HU interactions $\langle h_\mu h_\nu \rangle$ correlated strongly ($r_P = 0.73$, $P < 10^{-307}$) and had a low error ($\mathrm{nRMSE}_{\langle h_\mu h_\nu \rangle} = 0.17$). (**H**) The low-dimensional cRBM bottleneck reconstructs most neurons above chance level (purple), quantified by the normalized log-likelihood (nLLH) between neural test data $v_i$ and the reconstruction after being transformed to HU activity (see Materials and methods - 'Reconstruction quality'). Median normalized = $\mathrm{nLLH}_{\mathrm{cRBM}}$ 0.24. Reconstruction quality was also determined for a fully connected Generalized Linear Model (GLM) that attempted to reconstruct the activity of a neuron $v_i$ using all other neurons $\mathbf{v}_{-i}$ (see Materials and methods - 'Generalized Linear Model'). The distribution of 5000 randomly chosen neurons is shown (blue), with median $\mathrm{nLLH}_{\mathrm{GLM}} = 0.20$. The cRBM distribution is stochastically greater than the GLM distribution (one-sided Mann Whitney U test, $P < 10^{-42}$). (**I**) cRBM (purple) had a sparse weight distribution, but exhibited a greater proportion of large weights $w_{i,\mu}$ than PCA (yellow), both for positive and negative weights, displayed in log-probability. (**J**) Distribution of above-threshold absolute weights $|w_{i,\mu}|$ per neuron $v_i$ (dark purple), indicating that more neurons strongly connect to the cRBM hidden layer than expected by shuffling the weight matrix of the same cRBM (light purple). The threshold $\Theta$ was set such that the expected number of above-threshold weights per neuron $\mathbb{E}(\#\mathbf{w}_i > \Theta) = 1$. (**K**) Corresponding distribution as in (**J**) for PCA (dark yellow) and its shuffled weight matrix (light yellow), indicating a predominance of small weights in PCA for most neurons $v_i$. All panels of this figure show the data statistics of the cRBM with parameters $M = 200$ and $\lambda = 0.02$ (best choice after cross-validation, see *Figure 2—figure supplement 1*) of example fish #3, comparing the experimental test data test and model-generated data after cRBM training converged.

The online version of this article includes the following figure supplement(s) for figure 2:

**Figure supplement 1.** cRBM free parameter optimization by cross-validation.

**Figure supplement 2.** Generalized Linear Model (GLM) parameter optimization.

**Figure supplement 3.** cRBMs are in the compositional phase after convergence.

**Figure supplement 4.** Neurons can be embedded in multiple assemblies.

**Figure supplement 5.** Influence of number of HUs $M$ and sparsity regularization $\lambda$ on cRBM properties.

**Figure supplement 6.** cRBM assemblies are sparse and spatially localized.

**Figure supplement 7.** Variational Autoencoder (VAE) models do not fit the second-order statistics of the neural data.

Next, we assessed the reconstruction quality after neural data was compressed by the cRBM low-dimensional bottleneck. The reconstruction quality is defined as the log-likelihood of reconstructed neural data $\mathbf{v}_{\mathrm{recon}}$ (i.e. $\mathbf{v}$ that is first transformed to the low-dimensional $\mathbf{h}$, and then back again to the high-dimensional $\mathbf{v}_{\mathrm{recon}}$, see Materials and methods - 'Reconstruction quality'). This is important to prevent trivial, undesired solutions like $w_{i,\mu} = 0 \,\forall\, i, \mu$ which would directly lead to $\langle h_\mu \rangle_{P(\mathbf{v},\mathbf{h})} = \langle h_\mu \rangle_{\mathrm{data}} = 0$ (potentially because of strong sparsity regularization). *Figure 2H* shows the distribution of cRBM reconstruction quality of all neurons (in purple), quantified by the normalized log-likelihood (nLLH) such that 0 corresponds to an independent model ($P(v_i(t)) = \langle v_i \rangle$) and 1 corresponds to perfect reconstruction (non-normalized LLH = 0). For comparison, we also reconstructed the neural activity using a fully connected Generalized Linear Model (GLM, see Materials and methods - 'Generalized Linear Model' and *Figure 2*, *Figure 2—figure supplement 2H*, blue). The cRBM nLLH distribution is significantly greater than the GLM nLLH distribution (one-sided Mann Whitney U test, $P < 10^{-42}$), with medians $\mathrm{LLH}_{\mathrm{cRBM}} = 0.24$ and $\mathrm{LLH}_{\mathrm{GLM}} = 0.20$. Hence, projecting the neural data onto the low-dimensional representation of the HUs does not compromise the ability to explain the neural activity. In fact, reconstruction quality of the cRBM slightly outperforms the GLM, possibly due to the suppression of noise in the cRBM estimate. The optimal ($\lambda = 0.02, M = 200$) choice of free parameters was selected by cross-validating the median of the cRBM reconstruction quality, together with the normalized RMSE of the five previously described statistics (*Figure 2—figure supplement 1*).

Lastly, we confirmed that the cRBM indeed resides in the compositional phase, characterized by $1 \ll m(t) \ll M$ where $m(t)$ is the number of HUs active at time point $t$ (*Figure 2—figure supplement 3A*). This property is a consequence of the sparse weight matrix $\mathbf{W}$, indicated by its heavy-tail log-distribution (*Figure 2I*, purple). The compositional phase is the norm for the presently estimated cRBMs, evidenced by the distribution of median $m(t)$ values for all recordings (average $\frac{\mathrm{median}(m)}{M}$ is 0.26, see *Figure 2—figure supplement 3B*). Importantly, the sparse weight matrix does not automatically imply that only a small subset of neurons is connected to the cRBM hidden layer. We validated this by observing that more neurons strongly connect to the hidden layer than expected by shuffling

the weight matrix (*Figure 2J*). Further, we quantified the number of assemblies that each neuron was embedded in, which showed that increasing the embedding threshold did not notably affect the fraction of neurons embedded in at least 1 assembly (93–94%, see *Figure 2—figure supplement 4*). To assess the influence of $M$ and $\lambda$ on the inferred assemblies, we computed, for all cRBM models trained during the optimization of $M$ and $\lambda$, the distribution of assembly sizes (*Figure 2—figure supplement 5A-F*). We found that $M$ and $\lambda$ controlled the distribution of assembly sizes in a consistent manner: assembly size was a gradually decreasing function of both $M$ and $\lambda$ (two-way ANOVA, both $P < 10^{-3}$). Furthermore, for $M$ and $\lambda$ values close to the optimal parameter-setting ($M = 200$, $\lambda = 0.02$), the changes in assembly size were very small and gradual. This showcases the robustness of the cRBM to slight changes in parameter choice.

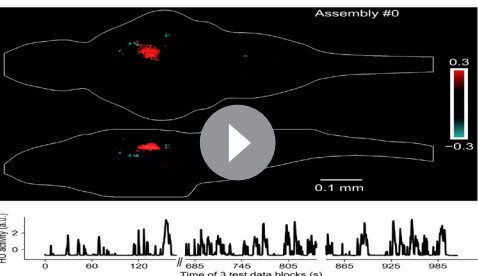

**Video 1.** All neural assemblies of one example fish All 200 inferred assemblies of the example fish #2 of *Figure 3* are shown in sequence. Top: neural assembly. Bottom: HU activity of test data.
https://elifesciences.org/articles/83139/figures#video1

Sparsity facilitated that each assembly only connects to a handful of anatomical regions, as we quantified by calculating the overlap between cRBM assemblies and anatomical regions (*Figure 2—figure supplement 6*). We found that cRBM assemblies connect to a median of three regions (interquartile range: 2–6 regions). Importantly, the cRBM has no information about the locations of neurons during training, so the localization to a limited set of anatomical areas that we observe is extracted from the neural co-activation properties alone. For comparison, Principal Component Analysis (PCA), a commonly used non-sparse dimensionality reduction method that shares the cRBM architecture, naturally converged to a non-sparse weight matrix (*Figure 2I*, yellow), with fewer connected neurons than expected by shuffling its weight matrix (*Figure 2K*). This led to unspecific assemblies that are difficult to interpret by anatomy (*Figure 2—figure supplement 6*). As a result, sparsity, a cRBM property shared with some other dimensionality reduction techniques, is crucial to interpret the assemblies by anatomy as we demonstrate in the next section.

We next asked whether sparsity alone was sufficient for a generative model to accurately recapitulate the neural recording statistics. To address this question, we trained sparse linear Variational Autoencoders (VAEs) using the same parameter-optimization protocol (*Figure 2—figure supplement 7A*). Like cRBMs, linear VAEs are generative models that learn a latent representation of a dataset (*Tubiana et al., 2019a*). We observed that VAEs were not able to replicate the second-order statistics, and therefore were not able to reconstruct neural activity from latent representation (*Figure 2—figure supplement 7B-D*), even though they also obtained sparse representations (*Figure 2—figure supplement 7E, F*). Other clustering or dimensionality reduction methods, such as k-means, PCA and non-negative matrix factorization, have been used previously to cluster neurons in the zebrafish brain (*Chen et al., 2018*; *Mu et al., 2019*; *Marques et al., 2020*). However, because these methods cannot generate artificial neural data using their inferred assemblies, their quality cannot be quantitatively assessed as we have done for the cRBM (but see *Tubiana et al., 2019a* for other comparisons).

## cRBM assemblies compose functional circuits and anatomical structures

Above, we have shown that cRBMs converge to sparse weight matrix solutions. This property enables us to visualize the cRBM-inferred neural assemblies as the collection of significantly connected neurons to an HU. Neurons from a given neural assembly display concerted dynamics, and so one may expect their spatial organization to reflect the neuroanatomy and functional organization of the brain. We here highlight a selection of salient examples of neural assemblies, illustrating that assemblies match well with anatomical structures and functional circuits, while the complete set of neural assemblies is presented in *Video 1*. In particular, we identified assemblies that together compose a neural circuit, are neurotransmitter-specific, encompass a long-range pathway, or can be identified by anatomy. The examples shown here are from a single fish (#3), but results from other fish were comparable.

First, we identified six assemblies that together span the hindbrain circuit that drives eye and tail movements (*Dunn et al., 2016*; *Wolf et al., 2017*; *Chen et al., 2018*). We find two neural assemblies in rhombomere 2 which align with the anterior rhombencephalic turning region (ARTR, *Ahrens et al., 2013*; *Dunn et al., 2016*; *Wolf et al., 2017*, *Figure 3A, B*). Each assembly primarily comprises neurons of either the left or right side of the ARTR, but also includes a small subset of contralateral neurons with weights of opposite sign in line with the established mutual inhibition between both subpopulations. Two other symmetric assemblies (*Figure 3C, D*) together encompass the oculomotor nucleus (nIII) and the contralateral abducens nucleus (nVI, in rhombomere 6), two regions engaged in ocular saccades (*Ma et al., 2014*) and under the control of the ARTR (*Wolf et al., 2017*). Additionally, we observed two symmetric assemblies (*Figure 3E, F*) in the posterior hindbrain (in rhombomere 7), in a region known to drive unilateral tail movements (*Chen et al., 2018*; *Marques et al., 2020*) and whose antiphasic activation is also controlled by the ARTR activity (*Dunn et al., 2016*).

Next, we observed assemblies that correspond to particular neurotransmitter expressions in the ZBrain atlas (*Randlett et al., 2015*), such as the excitatory Vglut2 (*Figure 3G*) and inhibitory Gad1b (*Figure 3H*) neurotransmitters. These assemblies consist of multiple dense loci that sparsely populate the entire brain, confirming that cRBMs are able to capture a large morphological diversity of neural assemblies. *Figure 3I* depicts another sparse, brain-wide assembly that encompasses the pallium, habenula (Hb) and interpeduncular nucleus (IPN), and thus captures the Hb-IPN pathway that connects to other regions such as the pallium (*Beretta et al., 2012*; *Bartoszek et al., 2021*).

Larger nuclei or circuits were often composed of a small number of distinct neural assemblies with some overlap. For example, the cerebellum was decomposed into multiple, bilateral assemblies (*Figure 3J*) whereas neurons in the torus semicircularis were grouped per brain hemisphere (*Figure 3K*). As a last example, the optic tectum was composed of a larger set of approximately 18 neural assemblies, which spatially tiled the volume of the optic tectum (*Figure 3L*). This particular organization is suggestive of spatially localized interactions within the optic tectum, and aligns with the morphology of previously inferred assemblies in this specific region (*Romano et al., 2015*; *Diana et al., 2019*; *Triplett et al., 2020*). However, *Figure 3* altogether demonstrates that the typical assembly morphology of the optic tectum identified by our and these previous analyses does not readily generalize to other brain regions, where a large range of different assembly morphologies compose neural circuits.

Overall, the clear alignment of cRBM-based neural assemblies with anatomical regions and circuits suggests that cRBMs are able to identify anatomical structures from dynamical activity alone, which enables them to break down the overall activity into parts that are interpretable by physiologists in the context of previous, more local studies.

## HU dynamics cluster into groups and display slower dynamics than neurons

HU activity, defined as the expected value of $P(\mathbf{h}|\mathbf{v})$ (*Equation 9*), exhibits a rich variety of dynamical patterns (*Figure 4A*). HUs can activate very transiently, slowly modulate their activity, or display periods of active and inactive states of comparable duration. *Figure 4B* highlights a few HU activity traces that illustrate this diversity of HU dynamics. The top three panels of *Figure 4B* show the dynamics of the assemblies of *Figure 3A-F* which encompass the ARTR hindbrain circuit that controls saccadic eye movements and directional tail flips. HUs 99 and 161 drive the left and right ARTR and display antiphasic activity with long dwell times of ~15s, in accordance with previous studies (*Ahrens et al., 2013*; *Dunn et al., 2016*; *Wolf et al., 2017*). HU 102 and 163 correspond to the oculomotor neurons in the nuclei nIII and nVI that together drive the horizontal saccades. Their temporal dynamics are locked to that of the ARTR units in line with the previously identified role of ARTR as a pacemaker for the eye saccades (*Wolf et al., 2017*). HUs 95 and 135, which drive directional tail flips, display transient activations that only occur when the ipsilateral ARTR-associated HU is active. This is consistent with the previous finding that the ARTR alternating activation pattern sets the orientation of successive tail flips accordingly (*Dunn et al., 2016*). The fourth panel shows the traces of the brain-wide assemblies of *Figure 3G, I*, displaying slow tonic modulation of their activity. Finally, the bottom panel, which corresponds to the collective dynamics of assembly 122 (*Figure 3H*), comprises short transient activity that likely corresponds to fictive swimming events.

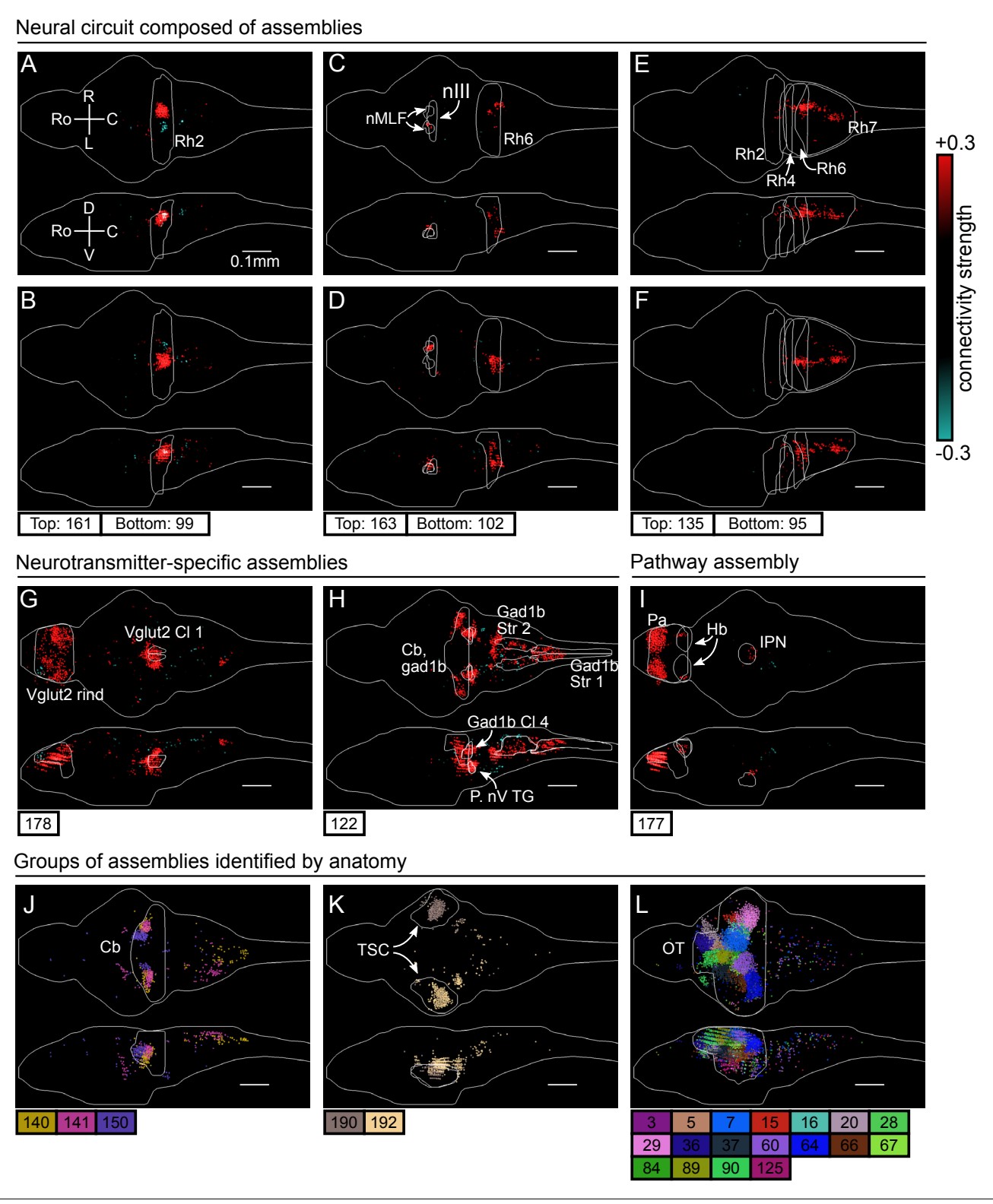

**Figure 3.** cRBM assemblies compose functional circuits and anatomical structures. (**A–I**) Individual example assemblies μ are shown by coloring each neuron $i$ with its connectivity weight value $w_{i,\mu}$ (see color bar at the right hand side). The assembly index μ is stated at the bottom of each panel. The orientation and scale are given in panel A (Ro: rostral, C: caudal, R: right, L: left, D: dorsal, V: ventral). Anatomical regions of interest, defined by the ZBrain Atlas (*Randlett et al., 2015*), are shown in each panel (Rh: rhombomere, nMLF: nucleus of the medial longitudinal fascicle; nIII: oculomotor nucleus nIII, Cl: cluster; Str: stripe, P. nV TG: Posterior cluster of nV trigeminal motorneurons; Pa: pallium; Hb: habenula; IPN: interpeduncular nucleus).

*Figure 3 continued on next page*

*Figure 3 continued*

(**J–L**) Groups of example assemblies that lie in the same anatomical region are shown for cerebellum (Cb), torus semicircularis (TSC), and optic tectum (OT). Neurons i were defined to be in an assembly μ when $|w_{i,\mu}| > 0.15$, and colored accordingly. If neurons were in multiple assemblies shown, they were colored according to their strongest-connecting assembly.

Some HUs regularly co-activate, leading to strong correlations between different HUs. This is quantified by their Pearson correlation matrix shown in *Figure 4C* (top), which reveals clusters of correlated HUs. These were grouped using hierarchical clustering (*Figure 4C*, bottom), and we then manually identified their main anatomical location (top labels). These clusters of HUs with strongly correlated activity suggest that much of the HU variance could be captured using only a small number of variables. We quantified this by performing PCA on the HU dynamics, finding that indeed 52% of the variance was captured by the first three PCs, and 85% by the first 20 PCs (*Figure 4D*). We further observed that HU activity is bimodal, as evidenced by the distribution of all HU activity traces in *Figure 4E*. This bimodality can emerge because the dReLU potentials $\mathcal{U}_\mu$ (*Equation 13*) can learn to take different shapes, including a double-well potential that leads to bimodal dynamics (see Materials and methods - 'Choice of HU potential'). This allows us to effectively describe HU activity as a two-state system, where $h_\mu(t) > 0$ increases the probability to spike ($P(v_i(t) = 1)$) for its positively connected neurons, and $h_\mu(t) < 0$ decreases their probability to spike. The binarized neuron activity is also a two-state system (spiking or not spiking), which enabled us to compare the time constants of neuron and HU state changes, quantified by the median time between successive onsets of activity. We find that HUs, which represent the concerted dynamics of neuronal assemblies, operate on a slower time scale than individual neurons (*Figure 4F*, *Figure 2—figure supplement 5G-L*). This observation aligns with the expected difference between cellular and circuit-level time scales.

## cRBM embodies functional connectivity that is strongly correlated across individuals

The probabilistic nature of cRBMs uniquely enables in silico perturbation experiments to estimate the functional connection $J_{ij}$ between pairs of neurons, where $J_{ij}$ is quantified by directly perturbing the activity of neuron $j$ and observing the change in probability to spike of neuron $i$. We first defined the generic, symmetric functional connection $J_{ij}$ using $P(v_i|v_j, v_{k\neq i,j})$ (*Equation 15*) and then used $P(\mathbf{v})$ (*Equation 12*) to derive the cRBM-specific $J_{ij}$ (*Equation 17*, see Materials and methods - 'Effective connectivity matrix'). Using this definition of $J_{ij}$, we constructed a full neuron-to-neuron effective connectivity matrix for each zebrafish recording. We then asked whether this cRBM-inferred connectivity matrix was robust across individuals. For this purpose, we calculated the functional connections between anatomical regions, given by the assemblies that occupy each region, because neuronal identities can vary across individual specimen. We aggregated neurons using the $L_1$ norm for each pair of anatomical regions to determine the functional connection between regions (see Materials and methods - 'From inter-neuron to inter-region connectivity'). For this purpose, we considered anatomical regions as defined by the mapzebrain atlas (*Kunst et al., 2019*) for which a regional-scale structural connectivity matrix exists to which we will compare our functional connectivity matrix.

This led to a symmetrical functional connectivity matrix for each animal, three of which are shown in *Figure 5A-C* (where non-imaged regions are left blank, and all eight animals are shown in *Figure 5—figure supplement 1*). The strength of functional connections is distributed approximately log-normal (*Figure 5D*), similar to the distribution of structural region-to-region connections (*Kunst et al., 2019*). To quantify the similarity between individual fish, we computed the Pearson correlation between each pair of fish. Functional connectivity matrices correlate strongly across individuals, with an average Pearson correlation of 0.69 (*Figure 5E and F*).

We conclude that similar functional circuits spontaneously activate across individuals, despite the limited duration of neural recordings (~25 minutes), which can be identified across fish using independently estimated cRBMs. In the next section, we aggregate these individual matrices to a general functional connectivity matrix for comparison with the zebrafish structural connectivity matrix.

## cRBM-inferred functional connectivity reflects structural connectivity

In the previous section we have determined the functional connections between anatomical regions using the cRBM assembly organization. Although functional connectivity stems from the structural

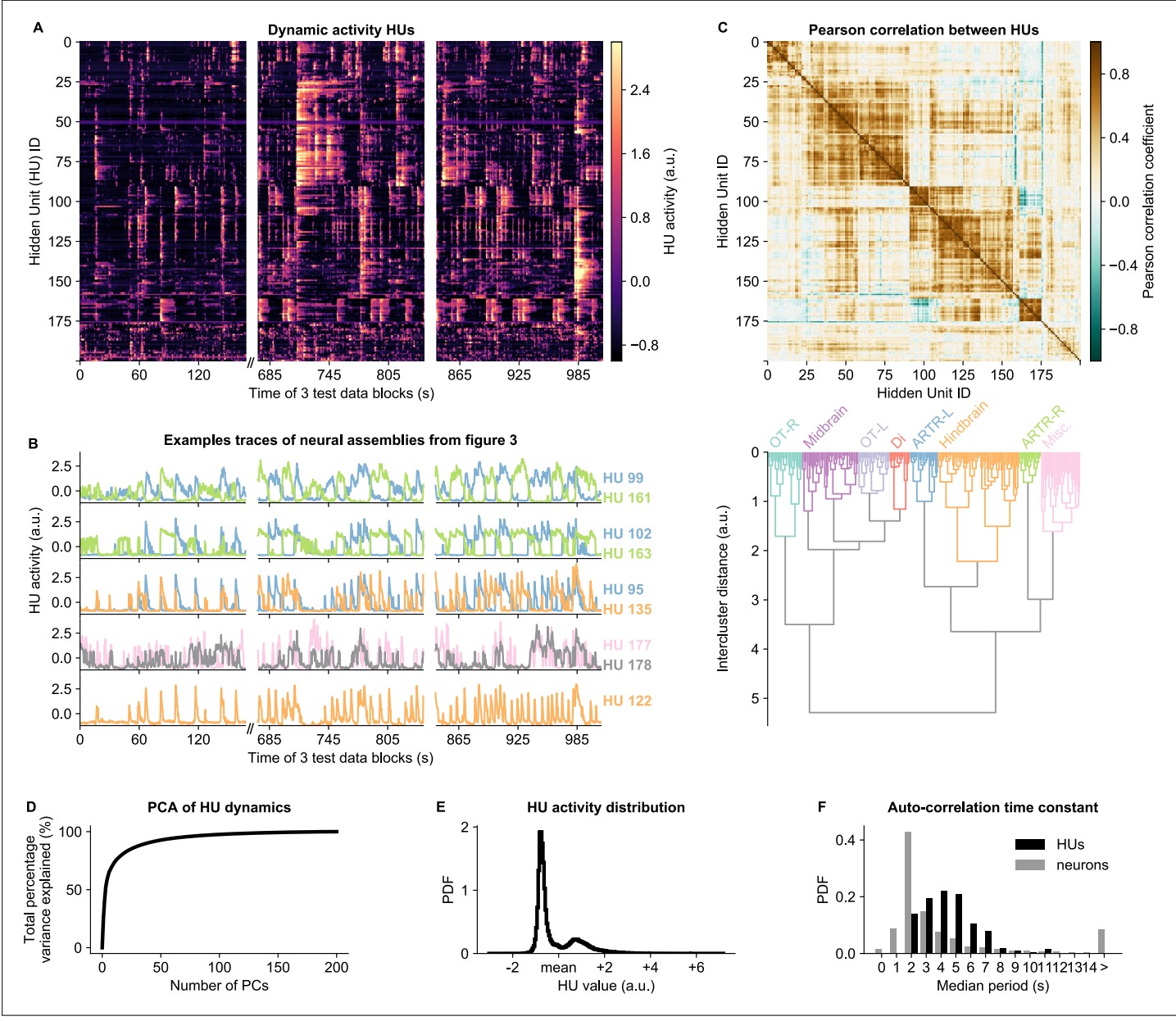

**Figure 4.** HU dynamics are bimodal and activate slower than neurons. (**A**) HU dynamics are diverse and are partially shared across HUs. The bimodality transition point of each HU was determined and subtracted individually, such that positive values correspond to HU activation (see Materials and methods - 'Time constant calculation'6.12). The test data consisted of three blocks, with a discontinuity in time between the first and second block (Materials and methods). (**B**) Highlighted example traces from panel A. HU indices are denoted on the right of each trace, colored according to their cluster from panel D. The corresponding cellular assemblies of these HU are shown in *Figure 3A-I*. (**C**) Top: Pearson correlation matrix of the dynamic activity of panel A. Bottom: Hierarchical clustering of the Pearson correlation matrix. Clusters (as defined by the colors) were annotated manually. This sorting of HUs is maintained throughout the manuscript. OT: Optic Tectum, Di: Diencephalon, ARTR: ARTR-related, Misc.: Miscellaneous, L: Left, R: Right. (**D**) A Principal Component Analysis (PCA) of the HU dynamics of panel A shows that much of the HU dynamics variance can be captured with a few PCs. The first 3 PCs captured 52%, the first 10 PCs captured 73% and the first 25 PCs captured 85% of the explained variance. (**E**) The distribution of all HU activity values of panel A shows that HU activity is bimodal and sparsely activated (because the positive peak is smaller than the negative peak). PDF: Probability Density Function. (**F**) Distribution of the time constants of HUs (black) and neurons (grey). Time constants are defined as the median oscillation period, for both HUs and neurons. An HU oscillation is defined as a consecutive negative and positive activity interval. A neuron oscillation is defined as a consecutive interspike-interval and spike-interval (which can last for multiple time steps, for example see *Figure 1A*). The time constant distribution of HUs is greater than the neuron distribution (Mann Whitney U test, $P < 10^{-16}$).

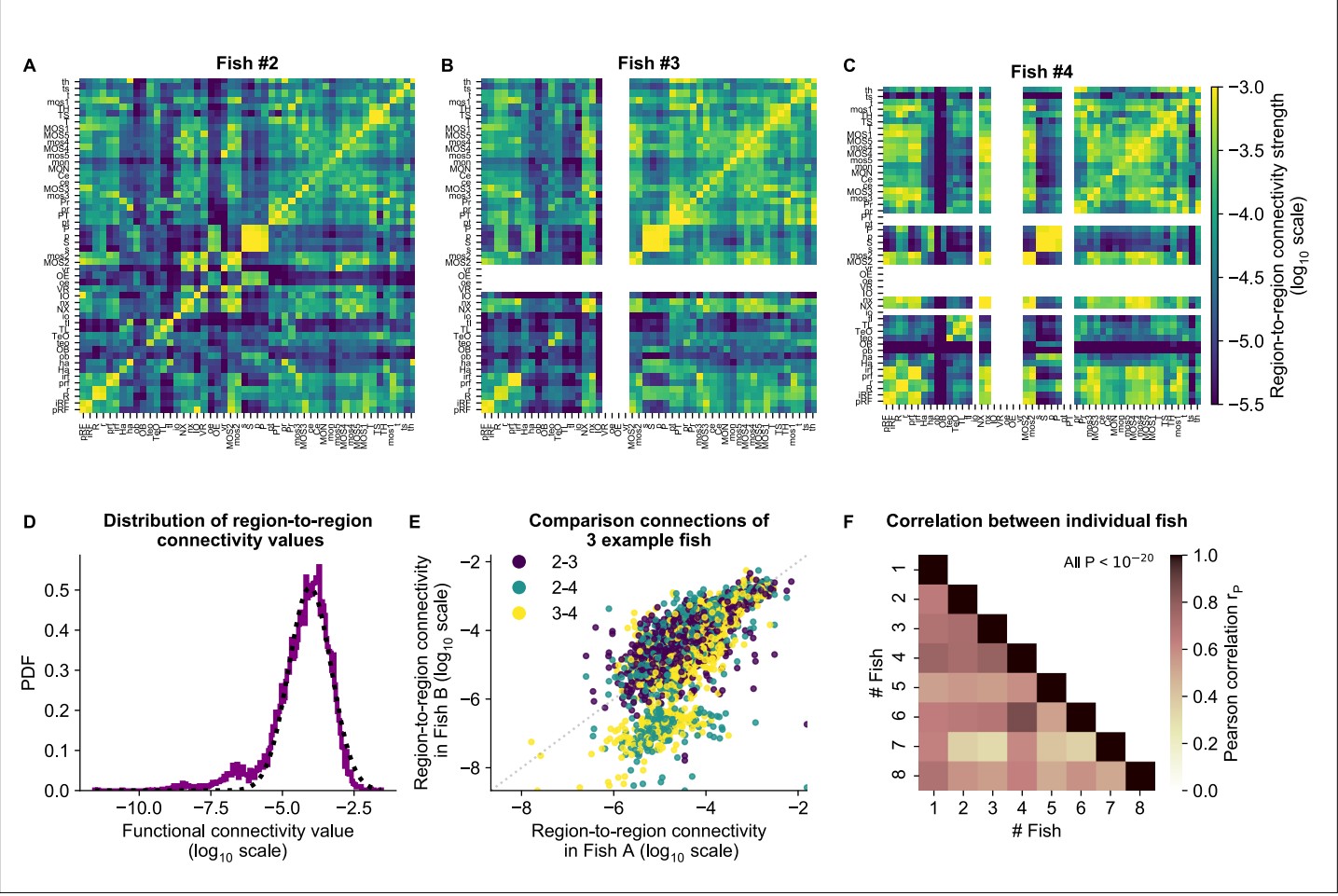

**Figure 5.** cRBM gives rise to functional connectivity that is strongly correlated across individuals. (**A**) The functional connectivity matrix between anatomical regions of the mapzebrain atlas (*Kunst et al., 2019*) of example fish #2 is shown. Functional connections between two anatomical regions were determined by the similarity of the HUs to which neurons from both regions connect to (Materials and methods). Mapzebrain atlas regions with less than five imaged neurons were excluded, yielding $N_{\mathrm{MAP}} = 50$ regions in total. See *Supplementary file 1* for region name abbreviations. The matrix is shown in $log_{10}$ scale, because functional connections are distributed approximately log-normal (see panel D). (**B**) Equivalent figure for example fish #3 (example fish of prior figures). (**C**) Equivalent figure for example fish #4. Panels A-C share the same $log_{10}$ color scale (right). (**D**) Functional connections are distributed approximately log-normal. (Mutual information with a log-normal fit (black dashed line) is 3.83, while the mutual information with a normal fit is 0.13). All connections of all eight fish are shown, in $log_{10}$ scale (purple). (**E**) Functional connections of different fish correlate well, exemplified by the three example fish of panels A-C. All non-zero functional connections (x-axis and y-axis) are shown, in $log_{10}$ scale. Pearson correlation $r_{\mathrm{P}}$ between pairs: $r_{\mathrm{P}}(\#2, \#3) = 0.73$, $r_{\mathrm{P}}(\#2, \#4) = 0.73$, $r_{\mathrm{P}}(\#3, \#4) = 0.78$. All correlation p values $< 10^{-20}$ (two-sided t-test). (**F**) Pearson correlations $r_{\mathrm{P}}$ of region-to-region functional connections between all pairs of 8 fish. For each pair, regions with less than five neurons in either fish were excluded. All p values $< 10^{-20}$ (two-sided t-test), and average correlation value is 0.69.

The online version of this article includes the following figure supplement(s) for figure 5:

**Figure supplement 1.** Functional connectivity matrices of all fish.

(i.e. biophysical) connections between neurons, it can reflect correlations that arise through indirect network interactions (*Bassett and Sporns, 2017*; *Das and Fiete, 2020*). Using recently published structural connectivity data of the zebrafish brain (*Kunst et al., 2019*), we are now able to quantify the overlap between a structurally defined connectivity matrix and our functional connectivity matrix estimated through neural dynamics. *Kunst et al., 2019* determined a zebrafish structural connectivity matrix between 72 anatomical regions using structural imaging data from thousands of individually Green Fluorescent Protein (GFP)-labeled neurons from multiple animals. We slightly extended this matrix by using the most recent data, filtering indirect connections and accounting for the resulting sampling bias (*Figure 6A*, regions that were not imaged in our light-sheet microscopy experiments

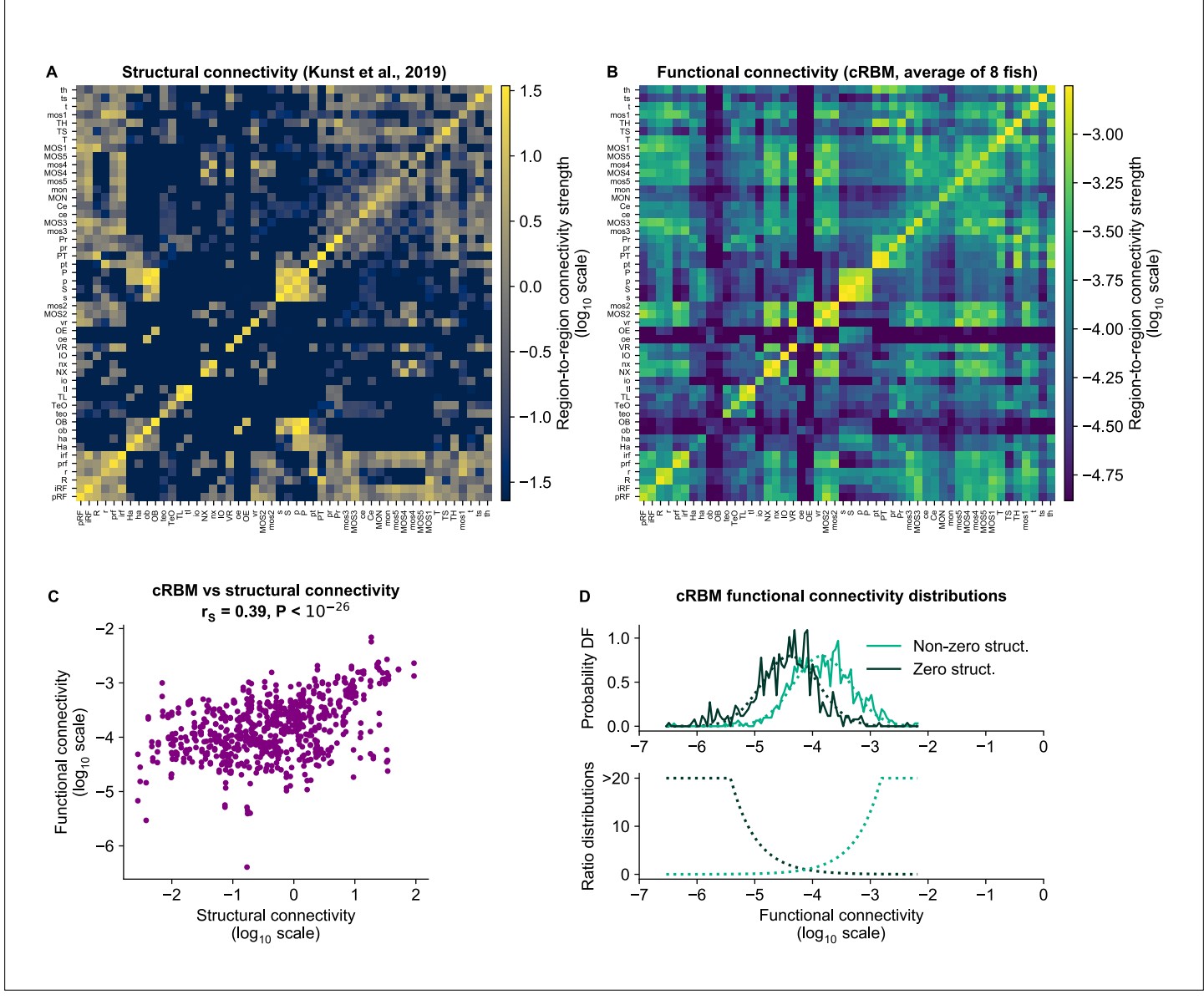

**Figure 6.** cRBM-inferred functional connectivity reflects structural connectivity. (**A**) Structural connectivity matrix is shown in $log_{10}$ scale, updated from Figure 8C of *Kunst et al., 2019*. Regions that were not imaged in our experiments were excluded (such that $N_{MAP} = 50$ out of 72 regions remain). Regions (x-axis and y-axis) were sorted according to *Kunst et al., 2019*. Compared to Figure 8C of *Kunst et al., 2019* additional structural data was added and the normalization procedure was updated to include within-region connectivity (see Materials and methods - 'Extensions of the structural connectivity matrix'). See *Supplementary file 1* for region name abbreviations. (**B**) Average functional connectivity matrix is shown in $log_{10}$ scale, as determined by averaging the cRBM functional connectivity matrices of all 8 fish (see Materials and methods - 'Specimen averaging of connectivity matrices'). The same regions (x-axis and y-axis) are shown as in panel A. (**C**) The average functional and structural connectivity of panels A and B correlate well, with Spearman correlation $r_S = 0.39$ ($P < 10^{-20}$, two-sided t-test). Each data point corresponds to one region-to-region pair. Data points for which the structural connection was exactly 0 were excluded (see panel D for their analysis). (**D**) The distribution of average functional connections of region pairs with non-zero structural connections is greater than functional connections corresponding to region pairs without structural connections ($P < 10^{-15}$, two-sided Kolmogorov-Smirnov test). The bottom panel shows the evidence for inferring either non-zero or zero structural connections, defined as the fraction between the PDFs of the top panel (fitted Gaussian distributions were used for denoising).

The online version of this article includes the following figure supplement(s) for figure 6:

**Figure supplement 1.** cRBM functional connectivity compared to baseline methods.

were excluded). Next, we aggregated the functional connectivity matrices of all our calcium imaging recordings to one grand average functional connectivity matrix (*Figure 6B*).

For comparison, we also calculated the connectivity matrices defined by either covariance or Pearson correlation (*Figure 6—figure supplement 1*). The cRBM functional connectivity spans a larger range of values than either of these methods, leading to a more fine-grained connectivity matrix akin to the structural connectivity map (*Figure 6B*). This greater visual resemblance was statistically confirmed by calculating the Spearman correlation between structural and functional connectivity, which is greater for cRBM ($r_S = 0.39$, *Figure 6C*), than for covariance-based connectivity ($r_S = 0.18$, *Figure 6—figure supplement 1* left) or correlation-based connectivity ($r_S = 0.26$, *Figure 6—figure supplement 1* right). Hence, using recordings of ~25 min on average, cRBMs were able to identify functional connections that resemble the anatomical connectivity between brain regions. Strong or weak functional connections are predictive of present or absent structural connections respectively (*Figure 6D*), and could thus potentially be used for inference in systems where the structural connectivity pattern is unknown.

## Discussion

We have developed a cRBM model that accurately replicated the data statistics of brain-scale zebrafish recordings, thereby forming neural assemblies that spanned the entire brain. The objective of our study was threefold: first, to show that the cRBM model can be applied to high-dimensional data, such as whole-brain recordings, second, to prove that an assembly-based model is sufficient to generate whole-brain neural data statistics, and third, to describe the physiological properties of the assembly organization in the zebrafish brain and use it to create a functional connectivity map. We have shown that, after convergence, the cRBM-generated data not only replicated the data statistics that it was constrained to fit, but also extrapolated to fit the pairwise correlation statistics of neurons and HUs, leading to a better reconstruction of neural data than a fully connected GLM (*Figure 2*). These results thereby quantify how neural assemblies play a major role in determining the collective dynamics of the brain. To achieve this, cRBMs formed sparsely localized assemblies that spanned the entire brain, facilitating their biological interpretation (*Figures 3 and 4*, *Figure 2—figure supplement 6*). Further, the probabilistic nature of the cRBM model allowed us to create a mesoscale functional connectivity map that was largely conserved across individual fish and correlated well with structural connectivity (*Figures 5 and 6*).

The maximum entropy principle underlying the cRBM definition has been a popular method for inferring pairwise effective connections between neurons or assemblies of co-activating cells (*Schneidman et al., 2006*; *Tavoni et al., 2017*; *Ferrari et al., 2017*; *Meshulam et al., 2017*; *Posani et al., 2018*; *Chen et al., 2019*). However, its computational cost has limited this pairwise connectivity analysis to typically $\sim 10^2$ neurons. The two-layer cRBM model that we used here alleviates this burden, because the large number of neuron-to-neuron connections are no longer explicitly optimized, which enables a fast data sampling procedure (*Figure 2B*). However, we have shown that these connections are still estimated indirectly with high accuracy via the assemblies they connect to (*Figure 2F*). We have thus shown that the cRBM is able to infer the $\frac{1}{2}N^2 \approx 10^9$ (symmetric) pairwise connections through its assembly structure, a feat that is computationally infeasible for many other methods. By implementing various algorithmic optimizations (Materials and methods - 'Algorithmic Implementation'), cRBM models converged in approximately 8–12 hr on high-end desktop computers (also see Materials and methods - 'Computational limitations').

Previously, we have extensively compared cRBM performance to other dimensionality reduction techniques, including Principal Component Analysis (PCA), Independent Component Analysis (ICA), Variational Autoencoders (VAEs) and their sparse variants, using protein sequence data as a benchmark (*Tubiana et al., 2019a*). Briefly put, we showed that PCA and ICA could not accurately model the system due to their deterministic nature, putting too much emphasis on low-probability high-variance states, while VAEs were unable to capture all features of data due to the unrealistic assumption of independent, Gaussian-distributed latent variables. In this study, we repeated this comparison with sparse linear VAEs, and reached similar conclusions: VAEs trained using the same protocol as cRBMs failed to reproduce second-order data statistics and to reconstruct neural activity via the latent layer, while the learnt assemblies were of substantially lower quality (indicated by a large fraction of disconnected HUs, as well as a highly

variable assembly size; *Figure 2—figure supplement 7*). Additionally, while PCA has previously been successful in describing zebrafish neural dynamics in terms of their main covariances modes (*Ahrens et al., 2012*; *Marques et al., 2020*), we show here that it is not appropriate for assembly extraction due to the absence of both a compositional and stochastic nature (*Figure 2*, *Figure 2—figure supplement 6*). Furthermore, we have shown that the generative component of cRBM models is essential for quantitatively assessing that the assembly organization is sufficient for reproducing neural statistics (*Figure 2*), moving beyond deterministic clustering analyses such as k-means (*Panier et al., 2013*; *Chen et al., 2018*), similarity graph clustering (*Mölter et al., 2018*) or non-negative matrix factorization (*Mu et al., 2019*) (see *Supplementary file 2*).

After having quantitatively validated the resultant assemblies, we moved to discussing the biological implications of our findings. Previous studies of the zebrafish optic tectum have identified neural assemblies that were spatially organized into single dense clusters of cells (*Romano et al., 2015*; *Diana et al., 2019*; *Triplett et al., 2020*). We have replicated these findings by observing the distinct organization of ball-shaped assemblies in the optic tectum (*Figure 3L*). However, our data extends to many other anatomical regions in the brain, where we found that assemblies can be much more dispersed, albeit still locally dense, consisting of multiple clusters of neurons (*Figure 3*). In sum, cRBM-inferred cell assemblies display many properties that one expects from physiological cell assemblies: they are anatomically localized, can overlap, encompass functionally identified neuronal circuits and underpin the collective neural dynamics (*Harris, 2005*; *Harris, 2012*; *Eichenbaum, 2018*). Yet, the cRBM bipartite architecture lacks many of the traits of neurophysiological circuits. In particular, cRBMs lack direct neuron-to-neuron connections, asymmetry in the connectivity weights and a hierarchical organization of functional dependencies beyond one hidden layer. Therefore, to what extent cRBM-inferred assemblies identify to neurophysiological cell assemblies, as postulated by *Hebb, 1949* and others, remains an open question.

cRBM allowed us to compute the effective, functional connections between each pair of neurons, aggregated to functional connections between each pair of regions, by perturbing neural activity in silico. Importantly, we found that this region-scale connectivity is well-conserved across specimen. This observation is non-trivial because each recording only lasted ~25 min, which represents a short trajectory across accessible brain states. It suggests that, although each individual brain may be unique at the neuronal scale, the functional organization could be highly stereotyped at a sufficiently coarse-grained level.

It would be naive to assume that these functional connections equate biophysical, structural connections (*Das and Fiete, 2020*). Both represent different, yet interdependent aspects of the brain organization. Indeed, we found that structural connectivity is well-correlated to functional connectivity, confirming that functional links are tied to the structural blueprint of brain connectivity (*Figure 6*). Furthermore, strong (weak) functional connections are predictive of present (absent) structural connections between brain regions, although intermediate values are ambiguous.

It will be crucial to synergistically merge structural and dynamic information of the brain to truly comprehend brain-wide functioning (*Bargmann and Marder, 2013*; *Kopell et al., 2014*). Small brain organisms are becoming an essential means to this end, providing access to a relatively large fraction of cells (*Ahrens and Engert, 2015*). To generate new scientific insights it is thus essential to develop analytical methods that can scale with the rapidly growing size of both structural and dynamic data (*Helmstaedter, 2015*; *Ahrens, 2019*). In this study, we have established that the cRBM can model high-dimensional data accurately, and that its application to zebrafish recordings was crucial to unveil their brain-scale assembly organization. In future studies, cRBMs could be used to generate artificial data whose statistics replicate those of the zebrafish brain. This could be used for further in silico ablation and perturbation studies with strong physiological footing, crucial for developing hypotheses for future experimental work (*Jazayeri and Afraz, 2017*; *Das and Fiete, 2020*). Lastly, the application of cRBMs is not specific to calcium imaging data, and can therefore be readily applied to high-dimensional neural data obtained by other recording techniques.

# Materials and methods

**Key resources table**

| Reagent type (species) or resource | Designation | Source or reference | Identifiers | Additional information |
|---|---|---|---|---|
| Software, algorithm | cRBM algorithm | This paper and *Tubiana and Monasson, 2017* | github.com/jertubiana/PGM | Materials and methods - 'Restricted Boltzmann Machines' , 'Compositional Restricted Boltzmann Machine' and Algorithmic Implementation |
| Software, algorithm | Fishualizer | *Migault et al., 2018* | bitbucket.org/benglitz/fishualizer_public | |
| Software, algorithm | Blind Sparse Deconvolution | *Tubiana et al., 2020* | github.com/jertubiana/BSD | |
| Software, algorithm | ZBrain Atlas | *Randlett et al., 2015* | engertlab.fas.harvard.edu/Z-Brain | |
| Software, algorithm | mapzebrain atlas | *Kunst et al., 2019* | fishatlas.neuro.mpg.de | |
| Software, algorithm | MATLAB (data preprocessing) | MathWorks | mathworks.com/products/matlab.html | |
| Software, algorithm | Computational Morphometry Toolkit (CMTK) | NITRC | nitrc.org/projects/cmtk | |
| Software, algorithm | Python | Python Software Foundation | python.org | |
| Strain, strain background (*Danio rerio, nacre mutant*) | Tg(elavl3:H2B-GCaMP6f) | *Quirin et al., 2016* | | |
| Strain, strain background (*Danio rerio, nacre mutant*) | Tg(elavl3:H2B-GCaMP6s) | *Vladimirov et al., 2014* | | |

## Data and code availability

The cRBM model has been developed in Python 3.7 and is available at: https://github.com/jertubiana/PGM, (copy archived at swh:1:rev:caf1d9fc545120f7f1bc1420135f980d5fd6c1fe; *Tubiana and van der Plas, 2023*). An extensive example notebook that implements this model is provided here.

Calcium imaging data pre-processing was performed in MATLAB (Mathworks) using previously published protocols and software (*Panier et al., 2013*; *Wolf et al., 2017*; *Migault et al., 2018*; *Tubiana et al., 2020*). The functional data recordings, the trained cRBM models and the structural and functional connectivity matrix are available at https://gin.g-node.org/vdplasthijs/cRBM_zebrafish_spontaneous_data.

Figures of neural assemblies or neurons (*Figures 1 and 3*) were made using the Fishualizer, which is a 4D (space +time) data visualization software package that we have previously published (*Migault et al., 2018*), available at https://bitbucket.org/benglitz/fishualizer_public. Minor updates were implemented to tailor the Fishualizer for viewing assemblies, which can be found at https://bitbucket.org/benglitz/fishualizer_public/src/assembly_viewer.

All other data analysis and visualization was performed in Python 3.7 using standard packages (numpy *Harris et al., 2020*), scipy (*Virtanen et al., 2020*), scikit-learn (*Pedregosa, 2011*), matplotlib (*Hunter, 2007*), pandas (*McKinney, 2010*), seaborn (*Waskom, 2021*), h5py. The corresponding code is available at https://github.com/vdplasthijs/zf-rbm (copy archived at swh:1:rev:b5df4e37434c0b18120485b8d856596db0b92444; *van der Plas, 2023*).

## Zebrafish larvae

Experiments were conducted on nacre mutants, aged 5–7 days post-fertilization (dpf). Larvae were reared in Petri dishes at 28 °C in embryo medium (E3) on a 14/10 hr light/dark cycle, and were fed powdered nursery food every day from 6 dpf. They were expressing either the calcium reporter GCaMP6s (fish 1–4, 6, and 8) or GCaMP6f (fish 5 and 7) under the control of the nearly pan-neuronal

promoter elavl3 expressed in the nucleus *Tg(elavl3:H2B-GCaMP6)*. Both lines were provided by Misha Ahrens and published by *Vladimirov et al., 2014* (H2B-GCaMP6s) and *Quirin et al., 2016* (H2B-GCaMP6f). Experiments were approved by Le Comité d'Éthique pour l'Experimentation Animale Charles Darwin C2EA-05 (02601.01).

## Light-sheet microscopy of zebrafish larvae

Spontaneous neural activity (i.e. in the absence of sensory stimulation) was recorded in larval zebrafish using light-sheet microscopy, which acquires brain-scale scans by imaging multiple $z$-planes sequentially (*Panier et al., 2013*; *Wolf et al., 2017*; *Migault et al., 2018*). Larvae were placed in 2% low melting point agarose (Sigma-Aldrich), drawn tail-first into a glass capillary tube with 1 mm inner diameter via a piston and placed in chamber filled with E3 in the microscope. Recordings were of length 1514 ± 238 seconds (mean ± standard deviation), with a brain volume imaging frequency of 3.9 ± 0.8 Hz.

The following imaging pre-processing steps were performed offline using MATLAB, in line with previously reported protocols (*Panier et al., 2013*; *Migault et al., 2018*). Automated cell segmentation was performed using a watershed algorithm (*Panier et al., 2013*; *Migault et al., 2018*) and fluorescence values of pixels belonging to the same neuron was averaged to obtain cell measurements. The fluorescence intensity values $F$ were normalized to $\Delta F/F = (F - \langle F \rangle)/(\langle F \rangle - F_0)$ where $\langle F \rangle$ is the baseline signal per neuron and $F_0$ is the overall background intensity (*Migault et al., 2018*). The $\Delta F/F$ activity of different imaging planes was subsequently temporally aligned using interpolation (because of the time delay between imaging planes; *Migault et al., 2018*) and deconvolved to binarized spike traces using Blind Sparse Deconvolution (BSD) (*Tubiana et al., 2020*). BSD estimates the most likely binary spike trace by minimizing the $L_2$ norm of the difference between the estimated spike trace convolved with an exponential kernel and the ground-truth calcium data, using $L_1$ sparsity regularization and online hyperparameter optimization. Calcium kernel time constants used for deconvolution were inferred using BSD on the spontaneous activity of three different fish (approximately 5000 neurons per fish, recorded at 10 Hz, previously reported by *Migault et al., 2018*). For the GCaMP6s line, we used a rise time of 0.2 s and a decay time of 3.55 s; for the GCaMP6f line, we used 0.15 s and 1.6 s, respectively.

Brain activity was recorded of 15 animals in total. Of these recordings, 1 was discarded because of poor image quality and 6 were discarded because neurons were inactive (defined by less than 0.02 spikes/(neurons × time points)), hence leaving 8 data sets for further analysis. The recorded brains were then registered onto the ZBrain Atlas (*Randlett et al., 2015*) and the mapzebrain atlas (*Kunst et al., 2019*) for anatomical labeling of neurons (*Migault et al., 2018*). The ZBrain Atlas was used in *Figures 1–4* because of its detailed region descriptions (outlining 294 regions in total). However, we also registered our data to the mapzebrain atlas (72 regions in total) in order to compare our results with the structural connectivity matrix which was defined for this atlas only (*Kunst et al., 2019*). Only neurons that were registered to at least 1 ZBrain region were used for analysis (to filter imaging artefacts). This resulted in $40709 \pm 13854$ neurons per recording (mean ± standard deviation, minimum = 23446, maximum = 65517).

## Maximum entropy principle

Here, we provide in brief the general derivation of the class of maximum entropy probabilistic models. Restricted Boltzmann Machines are an instance of this model, which is detailed in the following sections. The maximum entropy principle is used to create probabilistic models $P(x)$ (where $x$ denotes one data configuration sample) that replicate particular data statistics $f_k$, but are otherwise most random, and therefore least assumptive, by maximizing their entropy $H = -\sum_x P(x)\log(P(x))$(*Gardella et al., 2019*). The goal of the model is to match its model statistics $\langle f_k \rangle_{\text{model}} = \sum_x P(x)f_k(x)$ to the empirical data statistics $\langle f_k \rangle_{\text{data}} = F_k$. This is done using Lagrange multipliers $\Lambda_k$:

$$\widetilde{H} = -\sum_x P(x)\log\left(P(x)\right) - \sum_k \Lambda_k \left(\sum_x P(x)f_k(x) - F_k\right) \tag{5}$$

which yields, when $\widetilde{H}$ is maximized with respect to $P(x)$, the Boltzmann distribution (see, e.g., *Bialek, 2012* for a full derivation):

$$P(x) = \frac{1}{Z} \exp\left(-\ln 2 \sum_k \Lambda_k f_k(x)\right) = \frac{1}{Z} \exp\left(-E(x)\right) \qquad (6)$$

where $E(x)$ is defined as the resulting energy function. Importantly, the data dependency ($F_k$) disappears when going from *Equation 5* to *Equation 6*. Hence, the maximum entropy principle only defines the shape of the distribution $P(x)$, but not its specific parameters $\Lambda_k$ (*Bialek, 2012*). In the case of RBM, these are then optimized using maximum likelihood estimation, as detailed in the sections below.

## Motivation for choice of statistics

The derivation above describes the general maximum entropy model for a set of statistics $\{f_k\}$. The objective of this study is to extract the assembly structure from neural data, therefore creating two layers: a visible (neural data) layer $\mathbf{v} = (v_1, v_2, ..., v_N)$ and a hidden (latent) layer $\mathbf{h} = (h_1, h_2, ..., h_M)$. The model should capture the mean activity of each neuron $\langle v_i \rangle$, their pairwise correlations $\langle v_i v_j \rangle$, the neuron-HU interactions $\langle v_i h_\mu \rangle$ and a function of $h_\mu$. The latter is determined by the potential $\mathcal{U}$, which we set to be a double Rectified Linear Unit (dReLU), as motivated in the following sections. Fitting all $N^2$ pairwise interactions $\langle v_i v_j \rangle$ is computationally infeasible, but under the cell assembly hypothesis we assume that this should not be necessary because collective neural behavior is expected to be explained by membership to similar assemblies via $\langle v_i h_\mu \rangle$, and can therefore be excluded. We later show that pairwise correlations are indeed optimized implicitly (*Figure 2*). All other statistics are included and therefore explicitly optimized, also see *Equation 14*.

## Restricted Boltzmann machines

A Restricted Boltzmann Machine (RBM) is an undirected graphical model defined on a bipartite graph (*Smolensky, 1986*; *Hinton, 2002*; *Hinton and Salakhutdinov, 2006*), see *Figure 2A*. RBMs are constituted by two layers of random variables, neurons $\mathbf{v}$ and Hidden Units (HUs) $\mathbf{h}$, which are coupled by a weight matrix $\mathbf{W}$. There are no direct couplings between pairs of units within the same layer. Here, each visible unit $v_i$ corresponds to a single recorded neuron with binary (spike-deconvolved) activity ($v_i(t) \in \{0, 1\}$). Each Hidden Unit (HU) $h_\mu$ corresponds to the (weighted) activity of its neural assembly and is chosen to be real-valued. The joint probability distribution $P(\mathbf{v}, \mathbf{h})$ writes (*Hinton and Salakhutdinov, 2006*; *Tubiana and Monasson, 2017*):

$$P(\mathbf{v}, \mathbf{h}) = \frac{1}{Z} \exp\left(-E(\mathbf{v}, \mathbf{h})\right) = \frac{1}{Z} \exp\left(\sum_i g_i v_i - \sum_\mu \mathcal{U}_\mu(h_\mu) + \sum_{i,\mu} w_{i,\mu} v_i h_\mu\right) \qquad (7)$$

where $E$ is the energy function and $Z = \sum_\mathbf{v} \int_\mathbf{h} dv\, dh \cdot \exp\left(-E(\mathbf{v}, \mathbf{h})\right)$ is the partition function. The weights $g_i$ and potentials $\mathcal{U}_\mu$ control the activity level of the visible units and the marginal distributions of the HUs respectively, and the weights $w_{i,\mu}$ couple the two layers. Note that while $\mathbf{v}$ is directly observed from the neural recordings, $\mathbf{h}$ is by definition unobserved (i.e. hidden) and is sampled from the observed $\mathbf{v}$ values instead.

## From data to features

Given a visible layer configuration $\mathbf{v}$, a HU $h_\mu$ receives the input $I_\mu(\mathbf{v}) = \sum_i w_{i\mu} v_i \equiv \mathbf{w}_\mu^T \mathbf{v}$ and, owing to the bipartite architecture, the conditional distribution $P(\mathbf{h}|\mathbf{v})$ factorizes as:

$$P(\mathbf{h}|\mathbf{v}) = \prod_\mu P(h_\mu|\mathbf{v}) = \prod_\mu \exp\left(-\mathcal{U}_\mu(h_\mu) + h_\mu I_\mu(\mathbf{v}) - \Gamma_\mu(I_\mu(\mathbf{v}))\right) \qquad (8)$$

where $\Gamma_\mu(I) = \log\left(\int_h dh \cdot \exp\left(-\mathcal{U}_\mu(h) + hI\right)\right)$ is the cumulant generating function associated to the potential $\mathcal{U}_\mu$ that normalizes *Equation 8* (*Tubiana et al., 2019b*). The average activity of HU $h_\mu$ associated to a visible configuration $\mathbf{v}$ is given by a linear-nonlinear transformation (as defined by the properties of the cumulant generating function):

$$\langle h_\mu|\mathbf{v}\rangle = \frac{\partial \Gamma_\mu(I_\mu(\mathbf{v}))}{\partial I} = \Gamma'_\mu(\mathbf{w}_\mu^T \mathbf{v}) \qquad (9)$$

Throughout the manuscript, we use this definition to compute HU activity $h_\mu(t) = \langle h_\mu | \mathbf{v}(t) \rangle$ (e.g., in *Figure 4*).

## From features to data

Conversely, given a hidden layer configuration $\mathbf{h}$, a visible unit $v_i$ receives the input $I_i(\mathbf{h}) = \sum_\mu w_{i,\mu} h_\mu \equiv \mathbf{w_i}^T \mathbf{h}$ and the conditional distribution factorizes as:

$$P(\mathbf{v}|\mathbf{h}) = \prod_i P(v_i|\mathbf{h}) \propto \prod_i \exp\left(\left(g_i + I_i(\mathbf{h})\right)v_i\right) \tag{10}$$

and the average sampled $v_i$ activity is given by:

$$\langle v_i | \mathbf{h} \rangle = \sigma(\mathbf{w_i}^T \mathbf{h} + g_i) \tag{11}$$

where $\sigma(x) = 1/(1 + e^{-x})$ is the logistic function. Hence, a sampled visible layer configuration $\mathbf{v}$ is obtained by a weighted combination of the HU activity followed by Bernoulli sampling. RBMs are generative models, in the sense that they can generate new, artificial data using *Equations 8 and 10*. *Figure 2B* illustrates this Markov Chain Monte Carlo (MCMC) process, by recursively sampling from $P(\mathbf{h}|\mathbf{v})$ and $P(\mathbf{v}|\mathbf{h})$, which converges at equilibrium to $P(\mathbf{v}, \mathbf{h})$.

## Marginal distributions

The marginal distribution $P(\mathbf{v})$ has a closed-form expression because of the factorized conditional distribution of *Equation 9* (*Tubiana et al., 2019a*; *Tubiana et al., 2019b*):

$$P(\mathbf{v}) = \int \prod_{\mu=1}^{M} dh_\mu \cdot P(\mathbf{v}, \mathbf{h}) = \frac{1}{Z} \exp\left(\sum_{i=1}^{N} g_i v_i + \sum_{\mu=1}^{M} \Gamma_\mu(I_\mu(\mathbf{v}))\right) \tag{12}$$

For a quadratic potential $\mathcal{U}_\mu(h) = \frac{\gamma_\mu h_\mu^2}{2} + \theta_\mu h_\mu$, the cumulant generating function would also be quadratic and $P(\mathbf{v})$ would reduce to a Hopfield model, that is, a pairwise model with an interaction matrix $J_{ij} = \sum_\mu \frac{w_{i\mu} w_{j\mu}}{\gamma_\mu}$ (*Tubiana et al., 2019a*). Otherwise, $\Gamma_\mu$ is not quadratic, yielding high-order effective interaction terms between visible units and allowing RBMs to express more complex distributions. Importantly, the number of parameters remains limited, controlled by $M$ and does not scale as $N^2$ (unlike pairwise models).

## Choice of HU potential

The choice of HU potential determines three related properties: the HU conditional distribution $P(\mathbf{h}|\mathbf{v})$, the transfer function of the HUs and the parametric form of the marginal distribution $P(\mathbf{v})$. Hereafter we use the double-Rectified Linear Unit (dReLU) potential:

$$\mathcal{U}_\mu(h) = \frac{1}{2}\gamma_{\mu,+}h_+^2 + \frac{1}{2}\gamma_{\mu,-}h_-^2 + \theta_{\mu,+}h_+ + \theta_{\mu,-}h_-, \quad \text{where } h_+ = \max(h, 0), \ h_- = \min(h, 0) \tag{13}$$

Varying the parameters $\{\gamma_{\mu,+}, \gamma_{\mu,-}, \theta_{\mu,+}, \theta_{\mu,-}\}$ allows the potential to take a variety of shapes, including quadratic potentials ($\gamma_{\mu,+} = \gamma_{\mu,-}, \theta_{\mu,+} = \theta_{\mu,-}$), ReLU potentials ($\gamma_{\mu,-} \to \infty$) and double-well potentials (*Tubiana et al., 2019b*). The associated cumulant generating function $\Gamma(I)$ is non-quadratic in general, and depending on the parameters, the transfer function can be linear, ReLU-like (asymmetric slope and thresholding) or logistic-like (strong local slopes for binarizing inputs). Closed-form expressions of $\Gamma$ are detailed in *Tubiana et al., 2019a*; *Tubiana et al., 2019b*, and its derivatives are also detailed in *Tubiana, 2018*, p49-50. Note that the dReLU potential $\mathcal{U}_\mu$ and distribution $P(\mathbf{v})$ are invariant to the sign swap transformation $\gamma_{\mu,+}, \theta_{\mu,+} \Longleftrightarrow \gamma_{\mu,-}, \theta_{\mu,-}$ and $w_{i\mu} \Longleftrightarrow -w_{i\mu} \ \forall \ i, \mu$ (leading to $h_\mu \Longleftrightarrow -h_\mu$). For visual clarity, we perform this sign swap transformation after training on all HUs with predominantly negative weights (defined by $\sum_i w_{i,\mu} < 0$). Subsequently all HUs are positively activated if the group of neurons to which it connects is strongly active.

## RBM training

The RBM is trained by maximizing the average log-likelihood of the empirical data configurations $\mathcal{L} = \langle \log P(\mathbf{v}) \rangle_{\text{data}}$, using stochastic gradient descent methods. The gradient update steps are derived by calculating the derivative of $\mathcal{L}$, using *Equation 12*, with respect to the model parameters (*Tubiana et al., 2019a*):

$$
\begin{aligned}
\frac{\partial \mathcal{L}}{\partial g_i} &= \langle v_i \rangle_{\text{data}} - \langle v_i \rangle_{\text{model}} \\
\frac{\partial \mathcal{L}}{\partial w_{i\mu}} &= \langle v_i h_\mu \rangle_{\text{data}} - \langle v_i h_\mu \rangle_{\text{model}} \\
\frac{\partial \mathcal{L}}{\partial \theta_{\mu,\pm}} &= - \langle h_{\mu,\pm} \rangle_{\text{data}} + \langle h_{\mu,\pm} \rangle_{\text{model}} \\
\frac{\partial \mathcal{L}}{\partial \gamma_{\mu,\pm}} &= -\frac{1}{2} \left\langle h_{\mu,\pm}^2 \right\rangle_{\text{data}} + \frac{1}{2} \left\langle h_{\mu,\pm}^2 \right\rangle_{\text{model}}
\end{aligned}
\tag{14}
$$

Each gradient of $\mathcal{L}$ is thus the difference between a data statistic $\langle f_k \rangle_{\text{data}}$ and a model statistic $\langle f_k \rangle_{\text{model}}$. Hence the model learns to match these statistics to the training data. Importantly, model statistics $\langle f_k \rangle_{\text{model}}$ cannot be evaluated exactly due to the exponentially large number of data configurations (e.g. $2^N$ visible configurations). Therefore they are approximated by computing the statistics of model-generated data using the MCMC sampling scheme defined with *Equations 8 and 10* (see Materials and methods - 'Matching data statistics to model statistics' for more detail). MCMC sampling of a Boltzmann distribution in such high-dimensional space is in general very challenging owing to the exponentially long time to reach equilibrium. We use the persistent contrastive divergence approximation (*Tieleman, 2008*) and discuss its validity below.

## Compositional restricted boltzmann machine

In the previous sections, we have described the general properties of RBMs. We now motivate the specific RBM model choices that we have implemented, such as the dReLU potential and sparsity regularization, by discussing their impact on the properties of RBM-generated data.

Directed graphical models, for example, PCA, ICA, sparse dictionaries or variational autoencoders, prescribe a priori statistical constraints for their data representations, such as orthogonality/independence or specific marginal distributions such as Gaussian/sparse distributions. In contrast, the statistical properties of the representation of the data learned by RBMs are unknown a priori by construction (because of the maximum entropy principle). Instead, they emerge from the structure of the weight matrix, the potentials and the recursive back-and-forth sampling procedure described above. We have therefore previously studied the properties of typical samples of RBM with random weights as a function of the visible and hidden unit potentials and properties of the weight matrix using statistical mechanics tools (*Tubiana and Monasson, 2017*; *Tubiana et al., 2019a*). We have identified the three following typical behaviors, or phases.

In the ferromagnetic phase, a typical sample from $P(\mathbf{v}, \mathbf{h})$ has a single strongly activated HU ($m(t) \sim 1$, where $m(t)$ is the number of activated HUs at time $t$), whereas the others are not or merely weakly activated. The corresponding active visible units $v_i$ are defined by the weight vector $\mathbf{w}_{\mu^\star}$ associated to the active HU $h_{\mu^\star}$ (see *Equation 10*).

In the spin-glass phase, a typical sample does not have any relatively strongly activated HUs, but instead many moderately activated ones ($m(t) \sim M$). They interfere in a complex fashion to produce different visible unit configurations and there is no clear correspondence between the weight matrix and a typical data configuration.

Finally, in the compositional phase, a typical sample from $P(\mathbf{v}, \mathbf{h})$ has a small number of strongly activated HUs ($1 \ll m(t) \ll M$) whereas the others are weak or silent. Their weights are linearly combined through *Equation 10* to produce the corresponding visible layer configuration. The compositional phase is desirable because, firstly, there exists a simple link between the weight matrix and typical data configurations (they are obtained by combining a few weights), which facilitates interpretation of biological systems (*Tubiana et al., 2019b*). Secondly, the corresponding neural activity distribution is rich, as different choices of HU subsets yield a combinatorial diversity of visible layer configurations. Moreover, the modular nature of the compositional phase facilitates the assembly organization of neural dynamics, as motivated in the Introduction.

A set of sufficient conditions for the emergence of the compositional phase are (*Tubiana and Monasson, 2017*):

1. The HUs are unbounded and real-valued with a non-linear, ReLU-like transfer function.
2. The weight matrix $\mathbf{W}$ is sparse.
3. The columns $\mathbf{w}_\mu$ of the weight matrix have similar norm. (If a weight column associated to one HU is much larger than the others, visible configurations are solely aligned to it according to *Equation 10*.)

The first condition is satisfied by the dReLU potential (but not by quadratic potentials or binary-valued HUs). The second condition is enforced in practice by adding a $L_1$ sparse penalty term $\lambda \cdot \sum_{i\mu} |w_{i\mu}|$ to the log-likelihood cost function. In our experiments, the optimal sparsity parameter $\lambda$ was determined to be $\lambda = 0.02$ by cross-validation (*Figure 2—figure supplement 1*). The final condition is achieved by enforcing that $\mathrm{Var}(h_\mu) = 1$ and $\langle h_\mu \rangle \sim 0 \; \forall \; \mu$. This is done by an appropriate reparameterization of the HU potential of *Equation 13* and a batch-norm–like procedure, described in detail in *Tubiana, 2018*. This normalization promotes homogeneity among HU importance, preventing some units from being disconnected or others from dominating. In addition, ensuring that $h_\mu = \mathcal{O}(1)$ irrespective of the visible layer size (as opposed to e.g., $\frac{1}{2}(\gamma_+ + \gamma_-) = 1$ which yields $h_\mu \sim \sqrt{N}$) avoids the problem of ill-conditioned Hessians that was previously described by *Hinton, 2012*.

To emphasise the departure from the classic RBM formulation in this study, we name our model compositional RBM (cRBM).

## Algorithmic implementation

In the previous sections, we have described the cRBM model in full mathematical detail. The corresponding algorithmic implementation was adapted from *Tubiana et al., 2019b*. In addition, we have made several major implementation and algorithmic changes to accommodate the large data size of the zebrafish neural recordings. We provide the code open-source, and describe the code improvements and hyperparameter settings in this section. The following improvements were made, leading to a substantial reduction of computation time:

- Python 3 and numba (*Lam et al., 2015*) were used to compile custom functions, enabling SIMD vectorization and multicore parallelism.
- The sampling of $P(h_\mu | I_\mu)$ and evaluating its cumulant generating function $\Gamma_\mu$ and various moments requires repeated and costly evaluation of error functions erf and related functions (*Tubiana, 2018*, p49-50). Fast numerical approximations of these functions were implemented based on *Abramowitz et al., 1988* (p299).
- The number of memory allocation operations was minimized.
- The optimization algorithm was changed from stochastic gradient ascent to RMSprop (i.e. ADAM without momentum) with learning rate $5 \cdot 10^{-4}$ to $5 \cdot 10^{-3}$, $\beta_1 = 0$, $\beta_2 = 0.999$, $\epsilon = 10^{-6}$, see *Kingma and Ba, 2014* for a definition of the parameters. Compared to the original stochastic gradient ascent, the adaptive learning rates of RMSprop/ADAM yield larger updates for the weights attached to neurons with very sparse activity, resulting in substantially faster convergence.

### Hyperparameter settings

The following hyperparameters were used in the experiments of this manuscript:

- Number of hidden unit $M$: 200. This value was determined by cross-validation (*Figure 2—figure supplement 1*) on one data set (example fish #3). Because this cross-validation procedure was computationally expensive, the same value was used for all other data sets, except for 3 data sets which used $M = 100$ because their $N \approx \frac{1}{2} N_{\#3}$.
- Sparse regularization penalty $\lambda$: 0.02 (determined by cross-validation).
- Batch size: 100, 200, or 400. Larger batch sizes yield longer training time but more stable training; batch size was increased if training failed to converge.
- Number of Monte Carlo chains: 100.
- Number of gradient updates: $2 \cdot 10^5$.
- Number of Monte Carlo steps between each gradient update: 15.
- Initial learning rate $\eta$: between $5 \cdot 10^{-4}$ and $5 \cdot 10^{-3}$. We used $5 \cdot 10^{-3}$ by default and if weight divergence was observed, the learning was reinitialized with a reduced learning rate.

This occurred notably for high-$M$ and low-$\lambda$ models during the cross-validation procedure of *Figure 2—figure supplement 1*.

- Learning rate annealing scheme: the learning rate geometrically decayed during training, starting after 25% of the gradient update steps, from its initial value $\eta$ to a final value of $10^{-5}$.
- Number of training data samples: 70% of frames of each recording (=4086 training data samples on average), see section 'Train / test data split' for details.

## Computational limitations

We found that 57.5% ($= 23/40$) of cRBMs with optimal $(\lambda, M)$ settings successfully converged. cRBM models of these zebrafish data sets could be estimated in approximately 8–12 hr using 16 CPU threads (Intel Xeon Phi processor). The $(\lambda, M)$-cross-validation was therefore completed in three weeks using two desktop computers. Previously, we observed that this model requires a fixed number of gradient updates to converge, rather than a fixed number of epochs (*Tubiana et al., 2019a*; *Tubiana et al., 2019b*; *Bravi et al., 2021*). Hence, in principle, runtime does not strictly depend on the recording length, as the number of epochs can be reduced for longer recordings (assuming that the data distribution remains statistically stationary).

## Validity of the persistent contrastive divergence algorithm

Training RBM requires extensive MCMC sampling which is notoriously difficult for high-dimensional data sets. We resolve this by using Persistent Contrastive Divergence (PCD) to approximate the gradients (*Tieleman, 2008*). In this section, we discuss why this worked to successfully converge, despite the very large data size.

The typical number of Monte Carlo steps required to transition from one energy minimum to another through an energy barrier $\Delta E$ follows the Arrhenius law, scaling as $e^{\Delta E}$. In the thermodynamic limit ($N \to \infty$), $\Delta E$ scales as the system size $N$ multiplied by the typical energy required to flip a single visible unit, corresponding here to the inputs received from the hidden layer $I$. In contrast, for PCD only a limited number of MC steps (here, 15) are applied between each gradient update. Three factors explain why reasonably successful convergence was achieved in the trainings presented here.

Firstly, the use of the $L_1$ regularization limits the magnitude of the weights and therefore limits the input scale $I$. Secondly, in the compositional phase, the energy barriers do not scale as the full system size $N$ but rather as the size of one assembly $pN$ where $p$ is the fraction of non-zero weights (*Tubiana and Monasson, 2017*). Indeed, transitioning from one energy minimum, characterized by a subset of strongly activated HUs, to another minimum, characterized by another set of strongly activated HUs, is done by gradually morphing the first set into the second (*Roussel et al., 2021*). Compared to a direct transition, such a path is favored because the intermediate states are thermodynamically stable and energy barriers are smaller as each HU flip has an energy cost $\sim pN$. Lastly, throughout PCD training, MCMC sampling is not performed at thermal equilibrium and the model updates of the parameters of the distribution promote mixing (*Tieleman and Hinton, 2009*). This is seen from *Equation 14*: the log-likelihood gradient is the difference between the gradient of the energy averaged over the empirical data and the energy averaged over MCMC samples. Ascending the gradient amounts to pushing down the energy of data configurations and pushing up the energy of MCMC samples, thereby promoting mixing of the Markov chains.

Overall, combining small learning rates (and large number of gradient updates), large regularization, large number of Markov Chains and Monte Carlo steps has allowed convergence to be reached for the majority of cRBM training sessions.

## Functional connectivity inference

### Effective connectivity matrix

In this section, we present a derivation of the effective coupling matrix between neurons from the marginal distribution $P(\mathbf{v})$ using cRBMs. This is achieved by perturbing the activity of each neuron individually and quantifying the effect on other neurons. We first define the local coupling $J_{ij}$ between two neurons $v_i$ and $v_j$ for a generic probability distribution $P(v_1, v_2, \cdots, v_N)$, given a data configuration $\mathbf{v}$:

$$J_{ij}(\mathbf{v}) = \log\left(\frac{P(v_i = 1|v_1, \cdots, v_{i-1}, v_{i+1}, v_j = 1, \cdots, v_N)}{P(v_i = 1|v_1, \cdots, v_{i-1}, v_{i+1}, v_j = 0, \cdots, v_N)}\right) \\ - \log\left(\frac{P(v_i = 0|v_1, \cdots, v_{i-1}, v_{i+1}, v_j = 1, \cdots, v_N)}{P(v_i = 0|v_1, \cdots, v_{i-1}, v_{i+1}, v_j = 0, \cdots, v_N)}\right) \quad (15)$$

In other words, $J_{ij}$ is defined as the impact of the state of neuron $j$ on neuron $i$ in the context of activity pattern $\mathbf{v}$. Hence, the effective connectivity matrix $\mathbf{J}$ mathematically defines the functional connections, which can only be done using a probabilistic model $P(\mathbf{v})$. A positive (negative) coupling $J_{ij}$ indicates correlated (anti-correlated) collective behavior of neurons $i, j$. This effective coupling value is symmetric (because of Bayes' rule): $J_{ij}(\mathbf{v}) = J_{ji}(\mathbf{v})$. For context, note that $J_{ij}(\mathbf{v})$ is uniformly zero for an independent model of the form $P(v_1, \cdots, v_N) = \prod_i P_i(v_i)$, and that for a maximum entropy pairwise (Ising) model, with $P(v_1, \cdots, v_N) = \frac{1}{Z} \exp\left(\sum_i g_i v_i + \sum_{i<j} J_{ij}^{\text{ising}} v_i v_j\right)$, the $J_{ij}(\mathbf{v})$ matrix exactly identifies with the coupling matrix $J_{ij}^{\text{ising}}$, and does not depend on the data configuration $\mathbf{v}$ (so $J_{ij}(\mathbf{v}) = J_{ij}$).

However, in general, and for RBMs in particular, $J_{ij}(\mathbf{v})$ depends on the data set $\mathbf{v}$, and an overall coupling matrix can be derived by taking its average over all data configurations:

$$J_{ij} = \langle J_{ij}(\mathbf{v}) \rangle_{\text{data}} \quad (16)$$

Although *Equation 16* has a closed-form solution for RBMs (by inserting *Equation 12*), a naive evaluation requires $\mathcal{O}(N^3 MT)$ operations where $T$ is the number of data samples. However, a fast and intuitive approximation can be derived by performing a second-order Taylor expansion of $\Gamma_\mu(I_\mu)$:

$$J_{ij} = \sum_{\mu=1}^{M} w_{i\mu} w_{j\mu} \langle \Gamma_\mu''(\mathbf{v}) \rangle_{\text{data}} = \sum_{\mu=1}^{M} w_{i\mu} w_{j\mu} \langle \text{Var}\left(h_\mu|\mathbf{v}\right) \rangle_{\text{data}} \quad (17)$$

*Equation 17* is exact for quadratic potential and in general justified as the contribution of neurons $i, j$ is small compared to the scale of variation of $\Gamma_\mu$, $\mathcal{O}(\sqrt{pN})$ where $p$ is the fraction of non-zero couplings. In conclusion, we have mathematically derived the effective coupling between any two neurons $i$ and $j$. Intuitively, two neurons $i, j$ are effectively connected if they are connected to the same HUs (*Equation 17*).

## From inter-neuron to inter-region connectivity

In the above section, we have derived the inter-neuronal connectivity matrix $\mathbf{J}$. This matrix is then aggregated to an inter-regional connectivity matrix $\mathbf{J}^{\mathcal{R}}$ by taking the normalised $L_1$-norm of the corresponding $\mathbf{J}$ matrix block elements (i.e., $J_{km}^{\mathcal{R}} = \sum_{i \in R_k, j \in R_m} |J_{ij}|/(N_{R_k} \cdot N_{R_m})$, where $R_k$ is the set of neurons in region $k$).

Next, to derive the average connectivity matrix across multiple recordings, we used a weighted average of the individual recordings, with a region-pair specific weight equal to the length of the recording multiplied by the sum of the number of neurons in both regions (also see Section - 'Specimen averaging of connectivity matrices'). Compared to a naive average, this weighted average accounts for the variable number of neurons per region between recordings.

## Training cRBM models for connectivity estimates

Constructing the functional connectivity matrix of a cRBM does not require test data, but just the estimated weight matrix $\mathbf{W}$ (as explained above). Therefore we trained new cRBMs using the entire recordings (100% of data) to fully use the information available. cRBM training is stochastic, and to mitigate the possible variability that could arise we trained five cRBMs for each recording. Then, to assess convergence, we selected all cRBMs with $0.01 < \text{std}(\mathbf{w}) < 0.1$, where std denotes standard deviation, for further functional connectivity analysis (yielding 23 cRBMs for 8 data sets in total). Connectivity estimates of multiple cRBM models per data sets were averaged.

## Connectivity inference baselines

We considered four additional connectivity inference baseline methods:

- The covariance matrix.

- The Pearson correlation matrix.
- The sparse inverse covariance matrix inferred by graphical LASSO (*Friedman et al., 2008*) (as implemented in scikit-learn with default settings *Pedregosa, 2011*). Graphical LASSO is an efficient method for inference of large scale connectivity. Unfortunately, the implementation available failed to converge in reasonable time due to the high dimensionality of the data.
- The Ising model with pseudo-likelihood maximization (PLM) inference (*Ravikumar et al., 2010*).

Results obtained with the covariance and correlation matrices are presented in *Figure 6—figure supplement 1*. The connectivity matrices obtained by the PLM Ising model (not shown) correctly identified the diagonal entries of the region-region matrix, but not the off-diagonal coefficients and had a weaker correlation with the structural connectivity matrix than the covariance and correlation matrices ($r_S = 0.06$ using 4 fish).

## Optimizing the free parameters of cRBM

We set the free parameters $\lambda$ (sparsity regularization parameter) and $M$ (number of HUs) by cross-validating a large range of $(\lambda, M)$ values for one data set (fish #3). This was done by training cRBMs on 70% of the data, and evaluating model performance on the remaining test data, as detailed below. The resulting optimal values could then be used for all data sets (where $M$ was scaled with the number of neurons $N$). Importantly, the $(\lambda, M)$ parameters implicitly tune the average assembly size. Increasing the number of HUs and/or increasing the regularization strength decreases the average number of neurons per assembly (*Tubiana et al., 2019a*). Intuitively, assemblies that are too small do not have the capacity to capture high-order correlations, while assemblies that are too large would fail to account for local co-activations. Hence, the $(M, \lambda)$-cross-validation effectively identifies the optimal assembly sizes that fit the data statistics.

### Train / test data split

We split up one recording (fish #3) into training data (70% of recording) and withheld test data (30% of recording) for the free parameter $(\lambda, M)$ optimization procedure. This enabled us to assess whether the cRBMs learned to model the data statistics (as described in the main text, *Figure 2*, *Figure 2—figure supplement 1*), while ensuring that the cRBMs are not overfitted to the specific training data configurations. Importantly, this assumes that the test data comes from the same statistical distribution as the training data (while consisting of different data configurations). To ensure this, we split up the recording of example fish #3 (used for parameter optimization) in training and test splits as follows (before training the cRBMs): We divided the recording of length $T$ in 10 chronological segments of equal length (so that segment 1 has time points $\{t \in [1, \frac{T}{10})\}$ et cetera), with the rationale that by maintaining temporal order within each segment we would later be able to conduct dynamic activity analysis. This yielded $\binom{10}{3} = 120$ possible training/test splits of the neural data. We then evaluated the statistical similarity between the training and test split of each combination, by assessing the mean neural activity $\langle v_i \rangle$ and pairwise neural correlations $\langle v_i v_j \rangle - \langle v_i \rangle \langle v_j \rangle$ statistics. We quantified the similarity between training and test statistics by calculating the Root Mean Square Error ($\text{RMSE}(\mathbf{x}_1, \mathbf{x}_2) = \sqrt{\frac{1}{N_x} \sum_{n=1}^{N_x} \left( x_1(n) - x_2(n) \right)^2}$). The most similar split is defined by the lowest RMSE, but to show that cRBM are not dependent on picking the best possible split, but rather on avoiding the bad splits, we then chose to use the split with the $10^{\text{th}}$-percentile ranking RMSE. We hope that this aids future studies, where a potentially high number of possible splits prevents researchers from evaluating all possible splits, but a good split may nevertheless be found efficiently.

### Assessment of data statistics

Please note that the loss function, the log-likelihood, is computationally intractable and therefore cannot be readily used to monitor convergence or goodness-of-fit after training (*Fischer & Igel and Igel, 2012*). Moreover, approximations of the log-likelihood based on annealed importance sampling were found to be unreliable due to the large system size. However, because (c)RBM learn to match data statistics to model statistics (see Materials and methods–RBM training), we can directly compare these to assess model performance. Therefore, we assessed the following quantities.

## Matching data statistics to model statistics

Firstly, we evaluated three statistics that cRBMs are trained to optimize: the mean activity of neurons $\langle v_i \rangle$, the mean activity of HUs $\langle h_\mu \rangle$ and their pairwise interactions $\langle v_i h_\mu \rangle$. Additionally, second order statistics of pairwise neuron-neuron interactions $\langle v_i v_j \rangle$, HU-HU interactions $\langle h_\mu h_\nu \rangle$ and the reconstruction quality were evaluated, which the cRBM was not constrained to fit. Monitoring HU single and pairwise statistics $\langle h_\mu \rangle$ and $\langle h_\mu h_\nu \rangle$ served two purposes: (i) validation of model convergence and (ii) assessing whether correlations between assemblies can be captured by this bipartite model (i.e., without direct couplings between hidden units or an additional hidden layer). For each statistic $\langle f_k \rangle$, we computed its value based on empirical data $\langle f_k \rangle_{\text{data}}$ and on the model $\langle f_k \rangle_{\text{model}}$, which we then quantitatively compared to assess model performance.

Data statistics $\langle f_k \rangle_{\text{data}}$ were calculated on withheld test data (30% of recording). Naturally, the neural recordings consisted only of neural data $\mathbf{v}$ and not of HU data $\mathbf{h}$. We therefore computed the expected value of $\mathbf{h}_t$ at each time point $t$ conditioned on the empirical data $\mathbf{v}_t$, as further detailed in Methods - 'From data to features'.

Model statistics $\langle f_k \rangle_{\text{data}}$ cannot be calculated exactly, because that would require one to sample all possible states $P(\mathbf{v}, \mathbf{h})$, and were therefore approximated by evaluating cRBM-generated data. Here, 300 Monte Carlo chains were each initiated on random training data configurations and 50 configurations were sampled consecutively for each chain, with 20 sampling steps between saved configurations, after a burn-in period of 100 effective sampling configurations.

The $\langle v_i h_\mu \rangle$ statistic (**Figure 2C**) was corrected for the sparsity regularization, by adding the sparsity regularization parameter $\lambda$ to $\langle v_i h_\mu \rangle$ : $\langle v_i h_\mu \rangle_{\text{model}} = \langle v_i h_\mu \rangle_{\text{model-generated data}} + \lambda \cdot sign(w_{i,\mu})$. Furthermore, $(v_i, h_\mu)$ pairs with exactly $w_{i,\mu} = 0$ were excluded from analysis (5% of total for optimal cRBM in **Figure 2C**).

The pairwise neuron-neuron and HU-HU statistics ($\langle v_i v_j \rangle$, $\langle h_\mu h_\nu \rangle$) were corrected for their (trivially) expected correlation due to their mean activities (by subtraction of $\langle v_i \rangle \langle v_j \rangle$ and $\langle h_\mu \rangle \langle h_\nu \rangle$ respectively), so that only true correlations were assessed.

## Calculating the normalized Root Mean Square Error

Goodness of fit was quantified by computing the normalized Root Mean Square Error (nRMSE) for each statistic (shown in **Figure 2—figure supplement 1**). The RMSE between two vectors $\mathbf{x}_1, \mathbf{x}_2$ of length $N_x$ is defined as $\text{RMSE} = \sqrt{\frac{1}{N_x} \sum_{n=1}^{N_x} \left( x_1(n) - x_2(n) \right)^2}$. Ordinary RMSE was normalized so that different statistics could be compared, where 1 corresponds to $\text{nRMSE}_{\text{shuffled}}$, where both data and model statistics were randomly shuffled, and 0 corresponds to $\text{nRMSE}_{\text{optimal}}$ which is the RMSE between the training data and test data (by $\text{nRMSE} = 1 - \frac{\text{RMSE}_{\text{ordinary}} - \text{RMSE}_{\text{shuffled}}}{\text{RMSE}_{\text{optimal}} - \text{RMSE}_{\text{shuffled}}}$).

## Reconstruction quality

Additionally, we assessed the reconstruction quality of the test data. Here, the log-likelihood (LLH) between the test data $\mathbf{v}$ and its reconstruction $\mathbb{E}(\mathbf{v}_{\text{recon}}) = \mathbb{E}\left(\mathbf{v}|\mathbb{E}\left(\mathbf{h}|\mathbf{v}\right)\right) \in [0, 1]$ was computed. Because $v_i \in \{0, 1\}$, the LLH is defined as

$$\text{LLH}(v_i, \mathbb{E}(\mathbf{v}_{\text{recon}})) = \frac{1}{T} \sum_{t=1}^{T} \log \left( \mathbb{E}(\mathbf{v}_{\text{recon}})(t) * v_i(t) + (1 - \mathbb{E}\left(\mathbf{v}_{\text{recon}}\right)(t)) * \left(1 - v_i(t)\right) \right) \tag{18}$$

The resulting LLH was normalized (nLLH) such that 0 corresponds to an independent model (i.e., fitting neural activity with $\mathbb{E}(\mathbf{v}_{\text{recon}})(t) = \langle v_i \rangle \, \forall \, t$) and 1 to optimal performance (which is $\text{LLH}_{\text{optimal}} = 0$), by $\text{nLLH} = \frac{\text{LLH}_{\text{ordinary}} - \text{LLH}_{\text{independent}}}{-\text{LLH}_{\text{independent}}}$.

## Generalized Linear Model

We used logistic regression, a Generalized Linear Model (GLM), to quantify the reconstruction quality of a fully connected model (i.e., with neuron-to-neuron connections, see **Figure 2—figure supplement 2A**). Logistic regression makes a probabilistic binary prediction (**Bishop, 2006**), hence allowing direct comparison to the probabilistic estimates of neural activity by the cRBM. In logistic regression, for a neuron $v_i(t)$ at time $t$, the activity of all other neurons $\mathbf{v}_{-i}(t)$ at time $t$ was used to predict $\hat{v}_i(t) = P\left(v_i(t) = 1\right) = \frac{1}{1 + \exp\left(-\mathbf{w}_i \cdot \mathbf{v}_{-i}(t)\right)}$ where $\mathbf{w}_i$ is the estimated weight vector. This was implemented with scikit-learn (**Pedregosa, 2011**), using $L_2$ regularization. $L_2$ regularization was favored over $L_1$ as

it typically yields higher reconstruction performance; in the related context of protein contact map prediction, $L_2$-regularized GLMs also better reconstructed contacts than $L_1$-regularized GLMs (*Morcos et al., 2011*). The parameter $\lambda_{\mathrm{GLM}}$ was optimized to $\lambda_{\mathrm{GLM}} = 1000$ using cross-validation (*Figure 2—figure supplement 2B*). This is a computationally intensive model to compute because of the large number of regressor neurons $N - 1$: only $\sim 1000$ matrix rows could be inferred in 1 day on a 16-thread CPU desktop computer. Therefore, we performed the cross-validation of *Figure 2—figure supplement 2B* on 2% of all neurons (=1050 neurons) and computed the final distribution of *Figure 2H* on 10% of all neurons (=5252 neurons). GLMs were trained on the same train data as cRBMs, and evaluated on the same withheld test data as cRBMs (as described above).

## Variational Autoencoders

Variational Autoencoders (VAEs) were implemented in Tensorflow (2.1.10) using Keras (*Chollet, 2015*). For the encoder, we used a two-layer perceptron with intermediate layer size equal to the dimension of the latent space, a ReLU non-linearity for the intermediate layer and no non-linearity for the conditional mean and log-variance outputs. Batch normalization was used after each dense layer of the encoder. For the decoder, we used a dense layer with a sigmoid non-linearity (no batch normalization). To obtain sparse assemblies, a L1 penalty on the weights of the decoder was added, such that the latent variables correspond to sparse neural assemblies at generation time. Models were trained by ELBO maximization using a very similar protocol as cRBMs: 200 K updates using the Adam optimizer (initial learning rate $10^{-4}$, $\beta_1 = 0.9$, $\beta_2 = 0.999$ batch size: 100), with geometric decay schedule of learning rate after 50% of the training to a final learning rate of $10^{-5}$. We tested the same hyperparameter range as for cRBMs, and selected the optimal model based on the held-out ELBO values (10 Gaussian samples per data configuration were used to compute the ELBO). The optimal hyperparameters were $M = 300$, $\lambda_1 = 0.01$, but several values were very close to optimal (*Figure 2—figure supplement 7A*), including the value used for cRBMs ($M = 200$, $\lambda_1 = 0.02$). We chose the latter for comparison to cRBM for the sake of simplicity, although we also included performance metrics of the $M = 300, \lambda = 0.01$ VAE model (*Figure 2—figure supplement 7B-H*).

## Regional occupancy

We determined the anatomical region labels of each neuron by registering our recordings to the ZBrain Atlas (as described previously). This yields a matrix $L$ of size $N_{\mathrm{ZBA}} \times N$, which elements are $l_{r,i} = 1$ if neuron $i$ is embedded in region $r$ and 0 if it is not. A cRBM neural assembly of HU $\mu$ is defined by its weight vector $\mathbf{w}_\mu$ (of size $N$). Because cRBMs converge to sparse solutions, most of the weight elements will be very close to 0. To determine which anatomical regions are occupied by the assembly neurons with significantly nonzero weights, we computed the dot product between the weight vector $\mathbf{w}_\mu$ and matrix $L$, leading to a weighted region label vector (of size $N_{\mathrm{ZBA}}$) for each HU. The matrix of all $M$ weighted region label vectors is shown in *Figure 2—figure supplement 6A* for cRBM and *Figure 2—figure supplement 6B* for PCA.

The effective number of anatomical regions that one cRBM/PCA assembly is embedded in was then calculated using the Participation Ratio (PR) of each HU/Principal Axis. PRs are used to estimate the effective number of nonzero elements in a vector, without using a threshold (*Tubiana and Monasson, 2017*). The PR of a vector $\mathbf{x} = (x_1, \cdots, x_n)$ is defined by:

$$\mathrm{PR}(\mathbf{x}) = \frac{\left(\sum_{i=1}^{n} x_i^2\right)^2}{\sum_{i=1}^{n} x_i^4} \tag{19}$$

PR varies from $\frac{1}{n}$ when only 1 element of $\mathbf{x}$ is nonzero and $n$ when all elements are equal. We therefore estimated the effective number of regions by multiplying PR of the weighted region label vectors with the total number of regions $N_{\mathrm{ZBA}}$ in *Figure 2—figure supplement 6C*.

## Time constant calculation

The dReLU potential $\mathcal{U}_\mu$ of *Equation 13* can learn to take a variety of shapes, including a double-well potential (*Tubiana et al., 2019a*). HUs generally converged to this shape, giving rise to bimodal HU activity distributions (*Figure 4*). We determined the positions of the two peaks per HU using Gaussian Mixture Models fitted with two Gaussians. The bimodality transition point was then defined as the

average between the two peaks (which was approximately 0 for most HUs). To calculate the time constant of state changes between the two activity modes, we subtracted the bimodality transition point from each HU activity $h_\mu$ individually. For clarity, all dynamic activity traces shown (e.g. *Figure 4*) are thus bimodality transition point subtracted. The time constant of an activity trace was then defined as the period of a (two-state) oscillation. A HU oscillation is defined as a consecutive negative and positive activity interval (because the bimodality now occurs at 0). A neuron oscillation is defined as a consecutive interspike-interval and spike-interval (which can last for multiple time steps, for example see *Figure 1A*, right panel).

## Sorting of HUs

HUs were sorted by hierarchical clustering of the Pearson correlation matrix of their dynamic activity (*Figure 2B*). Hierarchical clustering was performed using the Ward variance minimization algorithm that defines the distance between clusters (*Virtanen et al., 2020*). This sorting of HUs (and thus assemblies) is maintained throughout the manuscript for the sake of consistency.

## Validating that the cRBM is in the compositional phase

To validate that the cRBMs converged to the compositional phase (see section - 'Compositional restricted boltzmann machine', compositional RBM formulation), we calculated the effective number of active HUs per data configuration (i.e., time step) $m(t) = \mathrm{PR}\left(\mathbf{h}_+(t)\right) \cdot M$ where PR is the participation ratio (*Equation 19*), $M$ the number of HUs and $h_+ = h - h_{\mathrm{inactive}}$, where $h_{\mathrm{inactive}}$ is the inactive peak as calculated with the Gaussian Mixture Models (see section - 'Time constant calculation'), because PR assumes that inactive elements are approximately zero (*Tubiana et al., 2019a*). A cRBM is said to be in the compositional phase if $1 \ll \mathrm{median}(m) \ll M$, which is true for all cRBMs (*Figure 2—figure supplement 3*).

## Extensions of the structural connectivity matrix

The inter-region structural connectivity matrix was derived from the single cell zebrafish brain atlas (*Kunst et al., 2019*). We used the post-publication updated data set from *Kunst et al., 2019* (timestamp: 28 October 2019). The data set consists of $N = 3098$ neurons, each characterized by the 3D coordinates of the soma center and of its neurites; there is no distinction between dendrites and axons. The brain is subdivided into $R = 72$ regions and each neuron is duplicated by left/right hemisphere symmetry. We aim to estimate $c_{r,r}$, the average strength of the connection between two neurons belonging to regions $r, r' \in [1, R]$. For each neuron $n \in [1, N]$, we determine, using region masks, the region $r(n)$ where its soma is located and the cumulative length of the intersection between all its neurites and each region $\ell_n(r)$. Under the assumptions that (i) the linear density of dendritic spines / axon presynaptic boutons is constant and (ii) the volumetric density of neurons is identical throughout regions, $L_n(r)$ is proportional to the volume $V_r$ of region $r$ times the average (bidirectional) connection strength between neuron $n$ and any neuron of region $r$. Aggregating over all neurons and symmetrizing, we obtain the following estimator for $c_{r,r}$:

$$c_{r,r'} = \mathrm{Symmetrized}\left\{ \frac{\sum_{n=1}^{N} \delta_{r(n),r} \times \ell_n(r')}{V_{r'} \times \sum_{n=1}^{N} \delta_{r(n),r}} \right\} \tag{20}$$

where $\delta_{r(n),r} = 1$ if neuron $n$ has its soma in region $r$ and 0 if not. Using the same notations, the formula previously used in *Kunst et al., 2019* is:

$$c_{r,r'} = \left\{ \frac{\sum_{n=1}^{N} \ell_n(r) + \ell_n(r')}{N(V_{r'} + V_r)} \right\} \tag{21}$$

*Equation 20* differs from *Equation 21* in three aspects:

1. It discriminates between direct and indirect connections. Previously, a structural connection between region $r$ and region $r'$ was established if a neuron had neurites with either tips or its soma within both regions. This may however result in indirect connections between $r$ and $r'$, in cases where the neuron soma resides in another region $r''$. Here, we only account for direct connections, resulting in an overall slightly sparser connectivity matrix.

2. It is well-defined along the diagonal, i.e., for intra-region connections, whereas in *Equation 21*, each neurite would be counted as a self-connection.
3. The denominator corrects for non-uniform sampling of the *traced* neurons throughout regions. Note that this issue only arose in the post-publication data set as non-uniform sampling was used to fill missing entries of the matrix.

## Specimen averaging of connectivity matrices

The number of neurons in a particular brain region can vary across recordings from different specimen. Since the entries of the connectivity matrix are expected to be more accurate for well-sampled regions, we computed the weighted average of region-to-region connections $c_{r,r'}$ as follows:

$$\langle c_{r,r'} \rangle = \frac{\sum_{\text{Fish F}} c_{r,r'}^F \cdot w_{r,r'}^F}{\sum_{\text{Fish F}} w_{r,r'}^F}$$
$$w_{r,r'}^F = T^F \frac{\left( N_{R_r}^F + N_{R_{r'}}^F \right)}{2} \tag{22}$$

Where $T^F$ is the recording length and $N_{R_r}^F$ is the number of neurons in region $r$ of fish $F$ that were recorded.

## Correlation analysis of connectivity matrices

Pearson correlation was used to assess the similarity between cRBM functional connectivity matrices of different individual animals (*Figure 5*). Spearman correlation was used to compare structural connectivity versus functional connectivity (*Figure 6*), because these two metrics do not necessarily scale linearly. All correlation analyses, and the Kilmogorov-Smirnov test of *Figure 6—figure supplement 1C*, performed on symmetric matrices excluded one off-diagonal triangle (of symmetrical values) to avoid duplicates.

## Acknowledgements

TLvdP had an Erasmus +fellowship (European Union) and acknowledges support from the Biotechnology and Biological Sciences Research Council (BBSRC, grant No. BB/M011224/1). JT acknowledges support from the Edmond J Safra Center for Bioinformatics at Tel Aviv University and from the Human Frontier Science Program (cross-disciplinary postdoctoral fellowship LT001058/2019 C). GM was funded by a PhD fellowship from the Doctoral School in Physics, Ile de France (EDPIF). GLG had a PhD fellowship from the Systems Biology Network of Sorbonne Université. BE and TLvdP. were supported by an NWO-VIDI Grant. Funding sources: European Research Council (ERC) under the European Union's Horizon 2020 research and innovation program grant agreement number 715980. Human Frontier Science Program (RGP0060/2017). The French Research National Agency under grant No. ANR-16-CE16-0017. Dutch Institute for Scientific Research NWO (Nederlandse Organisatie voor Wetenschappelijk Onderzoek) grant No. 016.VIDI.189.052. We thank the IBPS fish facility staff for the fish maintenance, in particular Stéphane Tronche and Alex Bois. We are grateful to Carounagarane Dore for his contribution to the design of the experimental setup. We thank Misha Ahrens for providing the GCaMP lines. We are also grateful to Rémi Monasson for very fruitful discussions and his comments on the manuscript.

## Additional information

### Funding

| Funder | Grant reference number | Author |
|---|---|---|
| Biotechnology and Biological Sciences Research Council | BB/M011224/1 | Thijs L van der Plas |

| Funder | Grant reference number | Author |
|---|---|---|
| Edmond J. Safra Center for Bioinformatics at Tel Aviv University | | Jérôme Tubiana |
| Human Frontier Science Program | LT001058/2019-C | Jérôme Tubiana |
| European Research Council | 715980 | Volker Bormuth |
| Human Frontier Science Program | RGP0060/2017 | Georges Debrégeas |
| Nederlandse Organisatie voor Wetenschappelijk Onderzoek | 016.VIDI.189.052 | Bernhard Englitz Thijs L van der Plas |

The funders had no role in study design, data collection and interpretation, or the decision to submit the work for publication.

## Author contributions

Thijs L van der Plas, Conceptualization, Data curation, Software, Formal analysis, Funding acquisition, Validation, Investigation, Visualization, Methodology, Writing – original draft, Writing – review and editing; Jérôme Tubiana, Conceptualization, Data curation, Software, Formal analysis, Funding acquisition, Validation, Investigation, Methodology, Writing – original draft, Writing – review and editing; Guillaume Le Goc, Geoffrey Migault, Data curation, Validation, Investigation, Writing – review and editing; Michael Kunst, Herwig Baier, Resources, Writing – review and editing; Volker Bormuth, Georges Debrégeas, Conceptualization, Resources, Data curation, Supervision, Funding acquisition, Investigation, Methodology, Project administration, Writing – review and editing; Bernhard Englitz, Conceptualization, Resources, Supervision, Funding acquisition, Investigation, Methodology, Project administration, Writing – review and editing

## Author ORCIDs

Thijs L van der Plas http://orcid.org/0000-0001-5490-1785
Jérôme Tubiana http://orcid.org/0000-0001-8878-5620
Guillaume Le Goc http://orcid.org/0000-0002-6946-1142
Herwig Baier http://orcid.org/0000-0002-7268-0469
Bernhard Englitz http://orcid.org/0000-0001-9106-0356
Georges Debrégeas http://orcid.org/0000-0003-3698-4497

## Ethics

Experiments were approved by Le Comité d'Éthique pour l'Experimentation Animale Charles Darwin C2EA-05 (02601.01).

## Decision letter and Author response

Decision letter https://doi.org/10.7554/eLife.83139.sa1
Author response https://doi.org/10.7554/eLife.83139.sa2

## Additional files

### Supplementary files

• Supplementary file 1. Table of abbreviations of mapzebrain atlas region names (used for interregional connectivity analyses).

• Supplementary file 2. Properties (relevant to this study) of common-used methods for analysing neural recordings. Abbreviations and example studies: Principal Component Analysis (PCA, *Ahrens et al., 2012*; *Lopes-dos-Santos et al., 2013*; *Marques et al., 2020*), Independent Component Analysis (ICA, *Lopes-dos-Santos et al., 2013*), k-means based algorithms (k-means, *Panier et al., 2013*; *Chen et al., 2018*; *Stringer et al., 2019*; *Bartoszek et al., 2021*), Non-Negative Matrix Factorization (NNMF, *Mu et al., 2019*), Variational Auto-Encoder (VAE, *Tubiana et al., 2019b*), Generalized Linear Model (GLM, *Bishop, 2006*), Boltzmann Machine (BM, *Schneidman et al., 2006*; *Meshulam et al., 2017*), compositional Restricted Boltzmann Machine (cRBM, this study).

Question marks denote tasks that are in principle feasible but computationally expensive and/or not demonstrated.

- MDAR checklist

## Data availability

The cRBM model has been developed in Python 3.7 and is available at: https://github.com/jertubiana/PGM (copy archived at swh:1:rev:caf1d9fc545120f7f1bc1420135f980d5fd6c1fe). An extensive example notebook that implements this model is also provided. Calcium imaging data pre-processing was performed in MATLAB (Mathworks) using previously published protocols and software (*Panier et al., 2013*; *Wolf et al., 2017*; *Migault et al., 2018*; *Tubiana et al., 2020*). The functional data recordings, the trained cRBM models and the structural and functional connectivity matrix are available at https://gin.g-node.org/vdplasthijs/cRBM_zebrafish_spontaneous_data. Figures of neural assemblies or neurons (Figure 1, 3) were made using the Fishualizer, which is a 4D (space + time) data visualization software package that we have previously published (Migault et al., 2018), available at https://bitbucket.org/benglitz/fishualizer_public . Minor updates were implemented to tailor the Fishualizer for viewing assemblies, which can be found at https://bitbucket.org/benglitz/fishualizer_public/src/assembly_viewer/. All other data analysis and visualization was performed in Python 3.7 using standard packages (numpy [*Harris et al., 2020*], scipy [*Virtanen et al., 2020*], scikit-learn [*Pedregosa, 2011*], matplotlib [*Hunter, 2007*], pandas [*McKinney, 2010*], seaborn [*Waskom, 2021*], h5py). The corresponding code is available at https://github.com/vdplasthijs/zf-rbm (copy archived at swh:1:rev:b5df4e37434c0b18120485b8d856596db0b92444).

The following dataset was generated:

| Author(s) | Year | Dataset title | Dataset URL | Database and Identifier |
|---|---|---|---|---|
| van der Plas TL, Tubiana J, Le Goc G, Migault G, Kunst M, Baier H, Bormuth V, Englitz B, Debregeas G | 2022 | Data from: Neural assemblies uncovered by generative modeling explain whole-brain activity statistics and reflect structural connectivity | https://gin.g-node.org/vdplasthijs/cRBM_zebrafish_spontaneous_data | GIN, cRBM_zebrafish_spontaneous_data |

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
