## [Editor Report]

Large scale recordings, sometimes involving 10s of thousands of neurons, are becoming increasingly common. Making sense of these recordings, however, is not easy. This paper introduces a new method, the compositional Restricted Boltzmann Machine, that overcomes this problem -- it can find structure in data, including both "cell assemblies" and structural connectivity, without inordinate computing resources (data from 40,000 neurons recorded from zebrafish can be analyzed in less than a day). This is a valuable contribution, both to those interested in data analysis, and to those interested in zebrafish.

---

## [Decision Letter]

[Editors' note: this paper was reviewed by Review Commons.]

---

## [Author Response]

Reviewer #1 (Evidence, reproducibility and clarity (Required)):Summary:In the present manuscript, van der Plas et al. compellingly illustrated a novel technique for engendering a wholebrain functional connectivity map from single-unit activities sampled through a large-scale neuroimaging technique. With some clever tweaks to the restricted Boltzmann Machine, the cRBM network is able to learn a low-dimensional representation of population activities, without relying on constrained priors found in some traditional methods. Notably, using some 200 hidden layer neurons, the employed model was able to capture the dynamics of over 40,000 simultaneously imaged neurons with a high degree of accuracy. The extracted features both illustrate the anatomical topography/connectivities and capture the temporal dynamics in the evolution of brain states. The illustrated technique has the potential for wide-spread applications spanning diverse recording techniques and animal species. Furthermore, the prospectives of modeling whole-brain network dynamics in 'neural trajectory' space and of generating artificial data in silico make for very enticing reasons to adopt cRBM.Major comments:1. Line 164. The authors claim that conventional methods "such as k-means, PCA and non-negative matrix factorization" cannot be quantitatively assessed for quality on the basis that they are unable to generate new artificial data. Though partly true, in most neuroscience applications, this is hardly cause for concern. Most dimensionality reduction methods (with few exceptions such as t-sne) allow new data points to be embedded into the reduced space. As such, quality of encoding can be assessed by cross-validation much in the same way as the authors described, and quantified using traditional metrics such as percentage explained variance. The authors should directly compare the performance of their proposed model against that of NNMF and variational auto-encoders. Doing so would offer a more compelling argument for the advantage of their proposed method over more widely-used methods in neuroscience applications. Furthermore, a direct comparison with rastermap, developed by Stringer lab at Janelia (https://github.com/MouseLand/rastermap), would be a nice addition. This method presents itself as a direct competitor to cRBM. Additionally, the use of GLM doesn't do complete justice to the comparison point used, since a smaller fraction of data were used for calculating performance using GLM, understandably due to its computationally intensive nature.

We thank the reviewer for the comment, and certainly agree that there are multiple methods for unsupervised feature extraction from data and that they can be validated for encoding quality by cross-validation. Below, we follow the reviewers suggestion to directly compare with VAEs, but argue first that a comparison with NNMF and rastermap is not appropriate. Specifically, we would like to stress that reconstructing through a low-dimensional, continuous bottleneck is a different (and arguably, easier) task, than generating whole distributions as the cRBM does. Reconstruction delineates the manifold of *possible* configurations, whereas generative modeling must *weigh* such configurations adequately. Moreover, none of the methodologies mentioned can perform the same tasks as cRBMs. For instance, NNMF learns localized assemblies, but cannot faithfully model inhibitory connections since, by definition, only nonnegative weights are learnt. Also, the connection between the learnt assemblies and the underlying connectivity is unclear. Similarly, rastermap is an algorithm for robustly i) sorting neurons along a set number of dimensions (typically 1 or 2) such that neighboring neurons are highly correlated, and ii) performing dimensionality reduction by clustering along these dimensions. Because Rastermap uses kmeans as the basis for grouping together neurons, it does not quantify connections between neurons and assemblies, nor assign neurons to multiple assemblies. Moreover, it is not a generative model, and thus cannot predict perturbation experiments, infer connectivities or assign probabilities to configurations. Therefore, we do not believe that NNMF or Rastermap would be a suitable alternative for cRBM in our study. We nonetheless appreciate the reviewer’s suggestions and agree that we should motivate more clearly why these methods are not applicable for our purposes. Therefore, to emphasize the relative merit of cRBM with respect to other unsupervised algorithms, we now provide a table (Supplementary Table 2) that lists their specific characteristics. We stress that we do not claim that cRBM are consistently better than these classical tools for dimensionality reduction, but focus only on the properties relevant to our study.

Regarding VAEs, we agree that these are close competitors of cRBMs, as they also jointly learn a representation and distribution of the data,. In *Tubiana et al. Neural Computation 2019,* we previously compared sparse VAEs with cRBMs for protein sequence modeling, and found that RBMs consistently outperformed VAEs. In the revised manuscript, we repeated the comparison with VAEs for Zebrafish neural recordings, and reached similar conclusions. Specifically, we found that for sparse linear VAEs trained using a similar protocol as cRBMs (ELBO loss minimization using ADAM optimizer, sparsity regularization, hyperparameter search using held-out validation set): i) the generated samples failed to replicate the second-order statistics of the data ii) the VAE could not reconstruct accurately neural spikes from the latent representation and iii) the majority (~60%) of the latent variables were completely disconnected from the neurons, and the remaining ones had highly variable size. This analysis shows that cRBMs consistently outperform VAE in terms of both interpretability and performance. The comparison between cRBM and VAE performance is now provided in the Manuscript (Section 7.10.4), and illustrated in Supplementary Figure S7 (shown below).

Results 2.2, last paragraph:

“We next asked whether sparsity alone was sufficient for a generative model to accurately recapitulate the neural recording statistics. To address this question, we trained sparse linear Variational Autoencoders (VAEs) using the same parameter-optimization protocol (Figure S7A). Like cRBMs, linear VAEs are generative models that learn a latent representation of a dataset (Tubiana et al., 2019a). We observed that VAEs were not able to replicate the second-order statistics, and therefore were not able to reconstruct neural activity from latent representation (Figure S7B-D), even though they also obtained sparse representations (Figure S7E, F).”

Discussion, 3rd paragraph

“In this study we repeated this comparison with sparse linear VAEs, and reached similar conclusions: VAEs trained using the same protocol as cRBMs failed to reproduce second-order data statistics and to reconstruct neural activity via the latent layer, while the learnt assemblies were of substantially lower quality (indicated by a large fraction of disconnected HUs, as well as a highly variable assembly size) (Figure S7).”

As for GLM, it is true that the comparison involved subsampling of the neurons (10%, i.e. 5252 neurons, due to the very high computational cost of GLM, where we could estimate the connectivity of ~1000 neurons per day). This was already denoted in the relevant figure caption, as the reviewer has seen, but we have now also clarified this point in Methods 7.10.3.

We think that, because this is a large and randomly selected sample of neurons, these 5252 neurons represent the full data set, and the GLM distribution of reconstruction likelihood (Figure 2H) will not change qualitatively if a larger sample would be used.

In contrast to GLM, optimized cRBM models converged learning of the entire data set in half a day, emphasizing their ability to handle very large datasets, such as the presently used zebrafish whole-brain recordings, which is crucial for any model to be applied in practice.

2. Line 26. The authors describe their model architecture as a formalization of cell assemblies. Cell assemblies, as originally formulated by Hebb, pertains to a set of neurons whose connectivity matrix is neither necessarily complete nor symmetric. Critically, in the physiological brain, the interactions between the individual neurons that are part of an assembly would occur over multiple orders of dependencies. In a restricted Boltzmann machine, neurons are not connected within the same layer. Instead, visible layer neurons are grouped into "assemblies" indirectly via a shared connection with a hidden layer neuron. Furthermore, a symmetrical weight matrix connects the bipartite graph, where no recurrent connectivities are made. As such, the proposed model still only elaborates symmetric connections pertaining to first-order interactions (as illustrated in Figure 4C). Such a network may not be likened with the concept of cell assemblies. The authors should refrain from detailing this analogy (of which there are multiple instances of throughout the text). It is true that many authors today refer to cell assemblies as any set of temporally-correlated neurons. However, saying "something could be a cell assembly" is not the same as saying "something is a cell assembly". How about sticking with cRBM-based cell assemblies (as used in section 2.3) and defining it beforehand?

We thank the reviewer for this excellent question. We agree that there is, in general, a discrepancy between computationally-defined assemblies and conceptual/neurophysiological definition of cell assemblies. We have added a clarification in Results 2.1 to clarify the use of this term when it first occurs in Results. However, we still believe that our work contributes to narrowing the gap. Indeed, our RBM-defined assemblies are i) localized, ii) overlapping, iii) rooted in connectivity patterns (both excitatory and inhibitory), and iv) cannot be reduced to a simple partitioning of the brain with full and uniform connectivity within and between partitions. This is unlike previous work based on clustering (no overlaps or heterogeneous weights), NNMF (no inhibition) or correlation network analysis (no low-dimensional representation).

Regarding the specific comments pointed here, we stress that:

– Effective interactions between neurons are not purely pairwise (“First order”), due to the usage of the non-quadratic potential. (see Equation 12-13). If the reviewer means by “First-order” interactions the lack of hierarchical organization, we agree, to some extent: in the current formulation, correlations between assemblies are mediated by overlaps between their weights. Fully hierarchical organization, e.g. by using Deep Boltzmann Machines or pairwise connections within the hidden layer is an interesting future direction, but on the other hand may make it hard to clearly identify assemblies as they might be spread out over multiple layers

– Neurons that participate in a given assembly (as defined by a specific hidden unit) are not all connected with one another with equal strength. Indeed, these neurons may participate in other assemblies, resulting in heterogeneity of connections (see Equation. 15-17) and interactions between assemblies.

– We acknowledge that the constraint of symmetrical connections is a core limitation of our method. Arguably, asymmetric connections are critical for predicting temporal evolution but less important for inferring a steady-state distribution from data, as we do here.

In the revised submission, we added a new paragraph in the Discussion section (lines 350-356) in which these limitations are discussed, including the imposed symmetry of the connections and the lack of hierarchical structures, copied below. We trust that this addresses the reviewer’s criticism:

In sum, cRBM-inferred cell assemblies display many properties that one expects from physiological cell assemblies: they are anatomically localized, can overlap, encompass functionally identified neuronal circuits and underpin the collective neural dynamics (Harris, 2005, 2012; Eichenbaum, 2018). Yet, the cRBM bipartite architecture lacks many of the traits of neurophysiological circuits. In particular, cRBMs lack direct neuron-to-neuron connections, asymmetry in the connectivity weights and a hierarchical organization of functional dependencies beyond one hidden layer. Therefore, to what extent cRBM-inferred assemblies identify to neurophysiological cell assemblies, as postulated by Hebb (1949) and others, remains an open question.

3. I would strongly recommend adding a paragraph discussing the limitation of using the cRBM, things future researchers need to keep in mind before using this method. One such recommendation is moving the runtimerelated discussion for cRBM, i.e. 8-12 hrs using 16 CPU from Methods to Discussion, since it's relevant for an algorithm like this. Additionally, a statement mentioning how this runtime will increase with the length of recordings and/or with the number of neurons might be helpful. What if the recordings were an hour-long rather than 25mins. This would help readers decide if they can easily use a method like this.

We thank the reviewer for the suggestion, and agree that it is important to cover the computational cost in the main text. Regarding the runtime for longer recordings, the general rule of thumb is that the model requires a fixed number of gradient updates to converge (20-80k depending on the data dimensionality) rather than a fixed number of epochs. Thus, runtime should not depend on recording length, as the number of epochs can be reduced for longer recordings. While we did not verify this rule for neural recordings, this is what we previously observed when modeling protein/DNA sequence data sets, whose size range from few hundreds to hundreds of thousands of samples (Tubiana et al., 2019, *eLife*; Tubiana et al. 2019, Neural Computation; Bravi et al. Cell Systems 2021; Bravi et al. PLOS CB 2021; Fernandez de Cossio Diaz et al. Arxiv 2022 Di Gioacchino et al. BiorXiv 2022). We have now added a summary of these points in Methods 7.7.2, also refer to this with explicit mention of the runtime in the Discussion, end of 2nd paragraph:

By implementing various algorithmic optimizations (Methods 7.7), cRBM models converged in approximately 812 hours on high-end desktop computers (also see Methods 7.7.2).

4. Line 515. A core feature of the proposed compositional RBM is the addition of a soft sparsity penalty over the weight matrix in the likelihood function. The authors claim that "directed graphical models" are limited by the a priori constraints that they impose on the data structure. Meanwhile, a more accurate statistical solution can be obtained using a RBM-based model, as outlined by the maximum entropy principle. The problem with this argument is that the maximum entropy principle no longer applies to the proposed model with the addition of the penalty term. In fact, the λ regularization term, which was estimated from a set of data statistics motivated by the experimenter's research goals (Figure S1), serves to constrict the prior probability. Moreover, in Figure S1F, we clearly see that reconstruction quality suffers with a higher penalty, suggesting that the principle had indeed been violated. That being said, RBMs are notoriously hard to train, possibly due to the unconstrained nature of the optimization. I believe that cRBM can help bring RBM into wider practical applications. The authors could test their model on a few values of the free parameter and report this as a supplementary. I believe that different parameters of λ could elaborate on different anatomical clusters and temporal dynamics. Readers who would like to implement this method for their own analysis would also benefit tremendously from an understanding of the effects of λ on the interpretation of their data. Item (1) on line 35 (and other instances throughout the text) should be corrected to reflect that cRBM replaces the hard constraints found in many popular methods with a soft penalty term, which allows for more accurate statistical models to be obtained.

We thank the reviewer for their analysis and suggestion. Indeed, adding the regularization term – not present in the classical formulation of the RBM (Hinton and Salakhutdinov, 2006, Science) – was critical for significantly enhancing its performance, which allowed us to implement this model on our large scale datasets (~50K visible units). We agree that providing more information on the effect of the regularization term will benefit readers who would like to use this method. We have now included an extensive analysis of the effects of λ in the revised manuscript, which is detailed in our answer to a similar question from reviewer 2. We therefore refer to our answer in response to the first question of reviewer 2.

The reviewer’s comment on the Maximum Entropy issue calls for some clarification. The maximum entropy principle is a recipe for finding the least constrained model that reproduces specified data-dependent moments. However, it cannot determine which moments are statistically meaningful in a finite-sized data set. A general practice is to only include low-order moments (1st and 2nd), but this is sometimes already too much for biological data. Regularization provides a practical means to select stable moments to be fitted and others to be ignored. This can be seen from the optimality condition, which writes, e.g., for the weights w_i,mu_:

| < v_i_ h_,mu_>data – < v_i_ h_,mu_>_model_ | < λ if w_i,mu_ = 0.

< v_i_ h_,mu_>_data_ – < v_i_ h_,mu_>_model_ | = λ(w_i,mu_) if |w_i,mu_| > 0.

Essentially, this lets the training decide which subset of the constraints should actually be used. Thus, regularized models are closer to the uniform distribution (g=w=0), and actually have higher entropy than unregularized one (see, e.g., Fanthomme et al. Journal of Statistical Mechanics, 2022). Therefore, we believe that a regularized maximum entropy model can still be considered a *bona fide* MaxEnt model. This formulation should not be confused with another formulation (that perhaps the reviewer has in mind) where a weighted sum of the entropy and the regularization term is maximized under the same moment-matching constraints. In this case, we agree that maximum entropy principle (MaxEnt) would be violated.

The choice of regularization value should be dictated by bias-variance trade-off considerations. Ideally, we would use the same criterion as for training, i.e., maximization of log-likelihood for the held-out test set, but it is intractable. Thus, we used a consensus between several tractable performance metrics as a surrogate; we believe this consensus to be principally independent of the research goal. While the reconstruction error indeed increases for large regularization values, this is simply because too few constraints are retained at high regularizations.

Essentially, the parameters selected by likelihood maximization find the finest assembly scale that can be accommodated by the data presented. Thus, the number and size of the assemblies are not specified by the complexity of the data set alone. Rather, the temporal resolution and length of the recordings play a key role; higher resolution recordings will allow the inference of a larger number of smaller assemblies, and enable the study of their hierarchical organization.

That being said, we fully agree that the regularization strength and number of hidden units have a strong impact on the nature of the representation learnt. In the revised manuscript, we follow the reviewer’s suggestion and provide additional insights on the effect of these parameters on the representation learnt (please see p9-10).

Minor comments:5. From a neuroscience point of view, it might be interesting to show what results are achieved using different values of M (say 100 or 300), rather than M=200, while still maintaining the compositional phase. Is there any similarity between the cRBM-based cell assemblies generated at different values of M? Is there a higher chance of capturing certain dynamics either functional or structural using cRBM? For example, did certain cRBM-based cell assemblies pop up more frequently than others at all values of M (100,200,300)?

Please find the answer p9-10, in response to a similar question raised by Reviewer #2, Q1.

6. The authors have mentioned that this approach can be readily applied to data obtained in other animal models and using different recording techniques. It might be nice to see a demonstration of that.

We agree that showing additional data analysis would be interesting, but we feel that it would overburden the supplementary section of the manuscript, which is already lengthy. In previous works, we and collaborators have used cRBMs for analyzing MNIST data (Tubiana and Monasson, 2017, PRL; Roussel et al. 2022 PRE), protein sequence data (Tubiana et al., 2019, *eLife*; Tubiana et al. 2019, Neural Computation; Bravi et al. Cell Systems 2021; Bravi et al. PLOS CB 2021; Fernandez de Cossio Diaz et al. Arxiv 2022), DNA sequences (Di Gioacchino et al. BiorXiv 2022), spin systems (Harsh et al. J. Phys. A 2020), etc. Many are included as example notebooks – next to the zebrafish data – in the linked code repository. For neural data, we have recently shared our code with another research group working on mice auditory cortex (2-photon, few thousands of neurons, Léger and Bourdieu). Preliminary results are encouraging, but not ready for publication yet.

7. Line 237. The justification for employing a dReLU transfer function as opposed to ReLU is unclear, at least within the context of neurobiology. Given that this gives rise to a bimodal distribution for the activity of HUs, the rationale should be clearly outlined to facilitate interpretability.

We thank the reviewer for the question. As we detail in the manuscript (Methods), the dReLU potential is one of the sufficient requirements for the RBM to achieve the compositional phase. The compositional phase is characterized by localized assemblies that co-activate to generate the whole-brain neural dynamics. This property reflects neurobiological systems (Harris, 2005, Neuron), which is one of the reasons why we employed compositional RBMs for this study.

As the reviewer points out, the HUs that we infer exhibit bimodal activity (Figure 4). Importantly, the HU activity is not constrained by the model to take this shape, as dReLU potentials allow for several activity distributions (see Methods 7.5.4; “Choice of HU potential”). In fact, ReLU potentials are a special case of dReLU (by (γ{∖mu, −}→∞)), so our model allows HU potentials to behave like ReLUs, but in practice they converge to a double-well potential for almost all HUs, leading to bimodal activity distributions.

Following the suggestion of the reviewer, we have now added this detail for clarity in Methods 7.5.4 and referenced this Methods section at line 237.

Reviewer #1 (Significance (Required)):van der Plas et al. highlighted a novel dimensionality reduction technique that can be used with success for discerning functional connectivities in large-scale single-unit recordings. The proposed model belongs to a large collection of dimensionality reduction techniques (for review, Cunningham, J., Yu, B. Dimensionality reduction for large-scale neural recordings. Nat Neurosci 17, 1500-1509 (2014). https://doi.org/10.1038/nn.3776; Paninski, L., and Cunningham, J. P. (2018). Neural data science: accelerating the experiment-analysis-theory cycle in largescale neuroscience. Current opinion in neurobiology, 50, 232-241.). The authors themselves highlighted some of the key methods, such as PCA, ICA, NNMF, variational auto-encoders, etc. The proposed cRBM model has also been published a few times by the same authors in previous works, although specifically pertaining to protein sequences. The use of RBM-like methods in uncovering functional connectivities is not novel either (see Hjelm RD, Calhoun VD, Salakhutdinov R, Allen EA, Adali T, Plis SM. Restricted Boltzmann machines for neuroimaging: an application in identifying intrinsic networks. Neuroimage. 2014 Aug 1;96:245-60. doi: 10.1016/j.neuroimage.2014.03.048.). However, given that the authors make a substantial improvement on the RBM network and have demonstrated the value of their model using physiological data, I believe that this paper would present itself as an attractive alternative to all readers who are seeking better solutions to interpret their data. However, as I mentioned in my comments, I would like to see more definitive evidence that the proposed solution has a serious advantage over other equivalent methods.Reviewer's expertise:This review was conducted jointly by three researchers whose combined expertise includes single-unit electrophysiology and two-photon calcium imaging, using which our lab studies the neurobiology of learning and memory and spatial navigation. We also have extensive experience in computational neuroscience, artificial neural network models, and machine learning methods for the analysis of neurobiological data. We are however limited in our knowledge of mathematics and engineering principles. Therefore, our combined expertise is insufficient to evaluate the correctness of the mathematical developments.Reviewer #2 (Evidence, reproducibility and clarity (Required)):In their manuscript, van der Plas et al. present a generative model of neuron-assembly interaction. The model is a restricted Boltzmann machine with its visible units corresponding to neurons and hidden units to neural assemblies. After fitting their model to whole-brain neural activity data from larval zebrafish, the authors demonstrate that their model is able to replicate several data statistics. In particular, it was able to replicate the pairwise correlations between neurons as well as assemblies that it was not trained on. Moreover, the model allows the authors to extract neural assemblies that govern the population activity and compose functional circuits and can be assigned to anatomical structures. Finally, the authors construct functional connectivity maps from their model that are then shown to correlate with established structural connectivity maps.Overall, the authors present convincing evidence for their claims. Furthermore, the authors state that their code to train their restricted Boltzmann machine models is already available on GitHub and that the data underlying the results presented in this manuscript will be made publicly available upon publication, which will allow people to reproduce the results and apply the methods to their data.One thing the authors could maybe discuss a bit more is the "right" parameter value M, especially since they used the optimal value of 200 found for one sample also for all the others. More specifically, how sensitive are the results to this value?

In the following we jointly address three of the reviewers’ questions (2 from reviewer 1, and 1 from reviewer 2).

Shortly summarized, the cRBM model has 2 free parameters; the number of hidden units M and the regularization parameter λ. In figures 2 and S1 we optimize their values through cross-validation, and then perform the subsequent analyses on models with these optimal values. The reviewers ask us to examine the outcome of the model for slightly different values of both parameters, in particular in relation to the sensitivity of the cRBM results to selecting the optimal parameters and the change in inferred assemblies and their dynamics.

We thank the reviewers for these questions and appreciate their curiosity to understand the effects of changing either of these two free parameters.

To assess the influence of both the number of hidden units M and the sparsity regularization parameter λ, we computed, for all cRBMs models trained in the hyperparameter search procedure, two metrics: the distribution of assembly sizes, and the distribution of HU dynamics time scales (as in Figure 4). We have now added a supplementary Figure S5 that shows these two distributions for the grid of M, λ values of Supplementary Figure S1, and have performed two-way ANOVA tests to determine the significance of M and λ on these two metrics. We found that M and λ controlled the distribution of assembly sizes in a consistent manner: assembly size was a gradually decreasing function of both M and λ (Figure S5A-F). Further, M, but not λ, similarly controlled the distribution of time scales (Figure S5G-L). However, for M and λ values close to the optimal parameter-setting (M=200, λ=0.01, determined by model selection), the changes in assembly size and time scale distributions were very gradual and minimal. This showcases the robustness of the cRBM to slight changes in parameter choice.

For illustration purposes, we also added panels M-O of the median-sized assemblies of 3 very different models (M=5, M=20, M=200).

Results 2.2:

To assess the influence of M and λ on the inferred assemblies, we computed, for all cRBM models trained during the optimization of M and λ, the distribution of assembly sizes (Figure S5 A-F). We found that *M* and λ controlled the distribution of assembly sizes in a consistent manner: assembly size was a gradually decreasing function of both M and λ(twoway ANOVA, both P < 10^-3^). Furthermore, for M and λ values close to the optimal parameter-setting (M=200, λ=0.02), the changes in assembly size were very small and gradual. This showcases the robustness of the cRBM to slight changes in parameter choice.

And, what happens if one would successively increase that number, would the number of assemblies (in the sense of hidden units that strongly couple to some of the visible units) eventually saturate?

This point will be addressed by inspecting models at different M values (see #1). We would like to further answer this question by referring to past work. In Tubiana et al., 2019, *eLife* (Appendix 1) we have done this analysis, and the result is consistent with the reviewer’s intuition. Because of the sparsity regularization, if M becomes larger than its optimum, the assemblies further sparsify without benefiting model performance, and eventually new assemblies duplicate previous assemblies or become totally sparse (i.e., all weights = 0) to not further induce a sparsity penalty in the loss function. So the ‘effective’ number of assemblies indeed saturates for high M.

Moreover, regarding the presentation, I have a few minor suggestions and comments that the authors also might want to consider:– In Figure 6C, instead of logarithmic axes, it might be better to put the logarithmic connectivity on a linear axis. This way the axes can be directly related to the colour bars in Figures 6A and B.

We agree and thank you for the suggestion. We have changed this accordingly (and also in the equivalent plots in figure S6).

– In Equation (8), instead of Γμ(I)∖\it should be Γμ(Iμ(v))

Done, thank you.

– In Section 7.0.5, it might make sense to have the subsection about the marginal distributions before the ones about the conditional distributions. The reason would be that if one wants to confirm Equation (8) one necessarily has to compute the marginal distribution in Equation (12) first.

We thank the reviewer for the suggestion, but respectfully propose to leave the section ordering as is. We understand what the reviewer means, but Equation (8) can also be obtained by factorizing P(v,h) Equation (7) and removing the v_i dependency. In Equation (8), \Γ can then be obtained by normalization. We believe this flow aligns better with the main text (where conditionals come first, when used for sampling, followed by the marginal of P(v) used for the functional connectivity inference).

– In Line 647f, the operation the authors are referring to is strictly speaking not an L1-norm of the matrix block. It might be better to refer to that e.g. as a normalised L1-norm of the matrix block elements.

Done, thank you.

– In Line 22, when mentioning dimensionality reduction methods to identify assemblies, it might make sense to also reference the work by Lopes-dos-Santos et al. (2013, J. Neurosci. Methods 220).

Done, thank you for the suggestion.

Reviewer #2 (Significance (Required)):The work presented in this manuscript is very interesting for two reasons. First, it has long been suggested that assemblies are a fundamental part of neural activity and this work seems to support that by showing that one can generate realistic whole-brain population activity imposing underlying assembly dynamics. Second, in recent years much work has been devoted to developing methods to find and extract neural assemblies from data and this work and the modelling approach can also be seen as a new method to achieve that. As such, I believe this work is relevant for anyone interested in neural population activity and specifically neural assemblies and certainly merits publication.Regarding my field of expertise, I used to work on data analysis of neural population activity and in particular on the question of how one can extract neural assemblies from data. I have to say that I have not much experience with fitting statistical models to data, so I can't provide any in-depth comments on that part of the work, although what has been done seems plausible.Reviewer #3 (Evidence, reproducibility and clarity (Required)):Summary: Understanding the organization of neural population activities in the brain is one of the most important questions in neuroscience. Recent technique advance has enabled researchers to record a large number of neurons and some times the whole brain. Interpreting and extracting meaningful insights from such data sets are challenging. van der Plas et al applied a generative model called compositional Restricted Boltzmann Machine (cRBM) to discover neuron assemblies from spontaneous activities of zebra fish brain. They found that neurons can be grouped into around 200 assemblies. Many of them have clear neurophysiological meaning, for example, they are anatomically localized and overlapped with known neural circuits. The authors also inferred a coarse-grained functional connectivity which is similar to known structural connectivity.The structure of the paper is well organized, the conclusion seems well supported by their numerical results. While this study provides a compelling demonstration that cRBM can be used to uncover meaningful structures from large neural recordings, the following issues limit my enthusiasm.Major:1) The overall implication is not clear to me. Although the authors mentioned this briefly in the discussion. It is not clear what else do we learn from discovered assemblies beyond stating that they are consist with previous study. For example, the author could have more analysis of the assembly dynamics, such as whether there are low dimensional structure etc.

First, we will comment on our analysis of the assemblies, before we continue to discuss the main implications of our work, which we believe are the inferred *generative* model of the zebrafish brain and the perturbation-based connectivity matrix that we discovered. Further, we have implemented the reviewer’s suggestion of analyzing the low-dimensional structure of the hidden unit activity, as further detailed in Question 7.

Indeed, the example assemblies that we show in Figure 3 have been thoroughly characterized in previous studies, which is why we chose to showcase these examples. Previous studies (including our own) typically focused on particular behaviors or sensory modalities, and aimed at identifying the involved neural circuit. Here, we demonstrate that by using cRBM on spontaneous activity recordings, one can simultaneously identify many of those circuits. In other words, these functional circuits/assemblies activate spontaneously, but in many different combinations and perhaps infrequently, so that it is very difficult to infer them from the full neural dynamics that they generate. cRBM has been able to do so, and Figure 3 (and supplementary video 1) serve to illustrate the variety of (known) circuits and assemblies that it inferred, some of which may represent true but not yet characterized circuits, which thus provide hypotheses for subsequent studies.

Further, we believe that the implication of our study goes beyond the properties of the assemblies we’ve identified, in several ways.

We demonstrate the power of cRBM’s generative capacity for inferring low-dimensional representations in neuroscience. Unlike standard dimensional reductionality methods, generative models can be assessed by comparing the statistics of experimental vs in-silico generated data. This is a powerful approach to validate a model, rarely used in neuroscience because of the scarcity of generative models compatible with large-scale data, and we hope that our study will inspire the use of this method in the field. We have made our cRBM code available, including notebook tutorials, to facilitate this.

The generative aspect of our model allowed us to predict the effect of single-neuron perturbations between all ~ 10^9^ pairs of neurons per fish, resulting in a functional connectivity matrix. We believe that the functional connectivity matrix is a major result for the field, similar to the structural connectivity matrix from Kunst et al., 2019, Neuron. The relation between functional and structural connectivity is unknown and of strong interest to the community (e.g., Das and Fiete, 2020, Nature Neuroscience). Our results allowed for a direct comparison of whole-brain region-by-region structural and functional connectivity. We were thus able to quantify the similarity between these two maps, and to identify specific region-pair matches and non-matches of functional and structural connectivity – which will be of particular interest to the zebrafish neuroscience community for developing future research questions.

Further, using these trained models – that will be made public upon publication – anyone can perform any type of *in silico* perturbation experiments, or generate endless artificial data with matching data statistics to the in vivo neural recordings.

We hope that this may convince the reviewer of the multiple directions of impact of our study. We will further address their comment on analysis of assembly dynamics below (question 7).

2) The learning algorithm of cRBM can be interpreted as matching certain statistics between the model and the experiment. For a general audience, it is not easy to understand ⟨hμ⟩data,⟨vihμ⟩data. Since these are not directly calculated from experimental observed activities v_i_, but rather the average is conditioned on the empirical distribution of p(v_i_). For example, the meaning of ⟨vihμ⟩data means ⟨vihi⟩data=1l∑v∈sEp(h|v)viμ, where S is the set of all observed neural activities: S={v1,⋯,vl}. The authors should explain this in the main text or method, since they are heavily loaded in figure 2.

We thank the reviewer for their suggestion, and have now implemented this. Their mathematics are correct; and we agree that it is not easy to understand without going through the full derivation of the (c)RBM. At the same time, we have tried not to alienate readers who might be more interested in the neuroscience findings than in understanding the computational method used. Therefore, we have kept mathematical details in the main text to a minimum (and have used schematics to indicate the statistics in Figures 2C-G), while explaining it in detail in Methods.

Accordingly, we have now extended section 7.10.2 (“Assessment of data statistics”) that explains how the data statistics were computed in Methods (and have referenced this in Results and in Methods 7.5.5), using the fact that we already explain the process of conditioning on **v** in Methods 7.5.1. The following sentences were added:

“[...]However, because (c)RBM learn to match data statistics to model statistics (see Methods 7.5.5), we can directly compare these to assess model performance. […]”

[..]

“For each statistic ⟨*fk*⟩ we computed its value based on empirical data ⟨*fk*⟩_data_ and on the model ⟨*fk*⟩_model_, which we then quantitatively compared to assess model performance. Data statistics ⟨*fk*⟩_data_ were calculated on withheld test data (30% of recording). Naturally, the neural recordings consisted only of neural data v and not of HU data **h**. We therefore computed the expected value of **h**_t_ at each time point t conditioned on the empirical data **v**_t_, as further detailed in Methods 7.5.1.”

[...]

3) As a modeling paper, it would be great to have some testable predictions.

We thank the reviewer for the enthusiasm and suggestion. We agree, and that is why we have included this in the form of functional connectivity matrices in Figures 5 and 6. To achieve this, we leveraged the generative aspect of the cRBM to perform *in silico* single-neuron perturbation experiments, which we aggregated to connectivity matrices. In other words, we have used our model to predict the functional connectivity between brain regions using the influence of single-neuron perturbations.

Obtaining a measure of functional connectivity/influence using single-neuron perturbations is also possible using state-of-the-art neuro-imaging experiments (e.g., Chettih and Harvey, 2019, Nature), though not at the scale of our *in silico* experiments. We therefore verify our predictions using structural data from Kunst et al., 2019, which we have extended substantially. We provide our functional connectivity result in full, and hope that this can inspire future zebrafish research by predicting which regions are functionally connected, which includes many pairs of regions that have not yet directly been studied in vivo.

Minor:1) The assembly is defined by the neurons that are strongly connected with a given hidden unit. Thus, some neurons may enter different assemblies. A statistics of such overlap would be helpful. For example, a Venn diagram in figure 1 that shows how many of them assigned to 1, 2, etc assemblies.

We thank the reviewer for this excellent suggestion. Indeed, neurons can be embedded in multiple assemblies. This is an important property of cRBMs, which deserves to be quantified in the manuscript. We have now added this analysis as a new supplementary figure 4. Neurons are embedded in an assembly if their connecting weight wi,μ is ‘significantly’ non-zero, depending on what threshold one uses. We have therefore shown this statistic for 3 values of the threshold (0.001, 0.01 and 0.1) – demonstrating that most neurons are strongly embedded in at least 1 assembly and that many neurons connect to more than 1 assembly.

Updated text in Results:

“Further, we quantified the number of assemblies that each neuron was embedded in, which showed that increasing the embedding threshold did not notably affect the fraction of neurons embedded in at least 1 assembly (93% to 94%, see Figure S4).”

2) What does the link between hidden units in Figure 1B right panel mean?

Thank you for the question, and we apologize for the confusion: if we understand the question right, the reviewer asks why the colored circles under the title ‘Neuronal assemblies of Hidden Units’ are linked. This schematic shows the same network of neurons as shown in gray at the left side of Figure 1B, but now colored by the assembly ‘membership’ of each neuron. Hence, the circles shown are still neurons (and not HUs), and their links still represent synaptic connections between neurons. We apologize for the confusion, and have updated the caption of Figure 1B to explain this better:

“[..] The neurons that connect to a given HU (and thus belong to the associated assembly), are depicted by the corresponding color labeling (right panel).[..]”.

3) A side-by-side comparison of neural activity predicted by model and the experimentally recorded activities would help the readers to appreciate the performance of the model. Such comparison can be done at both single neuron level or assembly level.

We thank the reviewer for this suggestion. The cRBM model is a statistical model, meaning that it fits the statistics of the data, and not the dynamics. The data that it generates therefore (should) adhere to the statistics of the training data, but does not reflect their dynamics. We believe that showing generated activity side-by-side of empirical activity is therefore not a meaningful example of generated data, as this would exemplify the dynamics, which this model is not designed to capture. Instead, in Figure 2, we show the statistics of the generated data versus the statistics of the empirical data (e.g., Figure 2C for the mean activity of all neurons). We believe that this is a better example representation of the generative performance of the model.

4) Definition of reconstruction quality in line 130.

We thank the reviewer for the suggestion, and have added the following sentence after line 130:

“The reconstruction quality is defined as the log-likelihood of reconstructed neural data v_recon_ (i.e., v that is first transformed to the low-dimensional h, and then back again to the high-dimensional v_recon_, see Methods 7.10.2).”

Further, please note that Methods describes the definition in detail (Eq 18 of the submitted manuscript), although we agree with the reviewer that more detail was required in the Results section at line 130.

5) Line 165. If PCA is compared with cRBM, why other dimensionality reduction methods, such as k-means and non-negative matrix factorization, can not be compared in terms of the sparsity?

Please see answer to question 1 from Reviewer 1.

6) Line 260, please provide minimum information about how the functional connectivity is defined based on assemblies discovered by cRBM.

We apologize if this was not clear. The first paragraph of this section (lines 248-259) of the submitted manuscript, provides the detail that the reviewer asks for, and we realize that the sentence of line 260 is better placed in the first paragraph, as it has come across as a very minimal explanation of how functional connectivity is defined.

We have now moved this sentence to the preceding paragraph, as well as specified the Method references (as suggested by this reviewer below), for additional clarity. We thank the reviewer for pointing out this sentence.

7) Some analysis of the hidden units population activities. Such as whether or not there are interesting low dimensional structure from figure 4A.

We thank the reviewer for their suggestion. In our manuscript we have used the cRBM model to create a lowdimensional (M=200) representation of zebrafish neural recordings (N=50,000). The richness of this model owes to possible overlaps between HUs/assemblies that can result in significant correlation in their activities. The latter is illustrated in Figure 4A-C: the activity of some HUs can be strongly correlated.

The reviewer’s suggestion is similar; to perform some form of dimensionality reduction on the low-dimensional HU activity shown in Figure 4. We have now added a PCA analysis to Figure 4 to quantify the degree of lowdimensional structure in the HU dynamics, and show the results in a new panel Figure 4D.

The following text has been added to the Results section:

These clusters of HUs with strongly correlated activity suggest that much of the HU variance could be captured using only a small number of variables. We quantified this by performing PCA on the HU dynamics, finding that indeed 52% of the variance was captured by the first 3 PCs, and 85% by the first 20 PCs (Figure 4D).

We believe that further visualization of these results, such as plotting the PC trajectories, would not further benefit the manuscript. The manuscript focuses on cRBM, and the assemblies/HUs it infers. Unlike PCA, these are not ranked/quantified by how much variance they explain individually, but rather they together ‘compose’ the entire system and explain its (co)variance (Figure 2). Breaking up a dominant activity mode (as found by PCA), such as the ARTR dynamics, into multiple HUs/assemblies, allows for some variation in activity of individual parts of the ARTR circuit (such as tail movement and eye movement generation), even though at most times the activity of these HUs is coordinated. We hope the reviewer agrees with our motivation to keep the manuscript focused on the nature of cRBM-inferred HUs.

8) Figure 4B right panel, how did the authors annotate the cluster manually? As certain assembly may overlap with several different brain regions, for example, figure 4D.

We thank the reviewer for this question, and we presume they meant to reference figure 3D as an example? For figure 4, as well as Figure 3, we used the ZBrain Atlas (Randlett et al., 2015) for definition of brain regions. This atlas presents a hierarchy of brain regions: for example, many brain regions are part of the rhombencephalon/hindbrain. This is what we used for midbrain/hindbrain/diencephalon. Further, many assemblies are solely confined to Optic Tectum (see Figure 3L), which we therefore used (split by hemisphere). Then, many brain regions are (partly) connected to the ARTR circuit, such as the example assembly of Figure 3D that the reviewer mentions. These we have all labeled as ARTR (left or right), though technically only part of their assembly *is* the ARTR. These two clusters therefore rather mean ‘ARTR-related’, in particular because their activity is locked to the rhythm of the ARTR (see Figure 4A). The final category is ‘miscellaneous’ (like Figure 3G).

However we agree that this wasn’t clear from the manuscript text, so we have changed the figure 4C caption to mention that ‘ARTR’ stands for ARTR-related assemblies, which we hope clarifies that ARTR-clustered assemblies can exist of multiple, disjoint groups of neurons, which relate to the ARTR circuit.

9) Better reference of the methods cited in the main text. The method part is quite long, it would be helpful to cite the section number when referring it in the main text.

We thank the reviewer for this helpful suggestion, we agree that it would benefit the manuscript to reference specific sections of the Methods. We have now changed all references to Methods to incorporate this.

10) Some discussion about the limitation of cRBM would be great.

We thank the reviewer for this suggestion, and have now included this. As Reviewer 1 had the same suggestion, we refer our answer to questions 2 and 3 from Reviewer 1 for more detail.

Reviewer #3 (Significance (Required)):This work provides a timely new technique to extract meaningful neural assemblies from large scale recordings. This study should be interested to both researchers doing either experiments and computation/theory. I am a computational neuroscientist.Description of analyses that authors prefer not to carry out.Please include a point-by-point response explaining why some of the requested data or additional analyses might not be necessary or cannot be provided within the scope of a revision. This can be due to time or resource limitations or in case of disagreement about the necessity of such additional data given the scope of the study. Please leave empty if not applicable.

We propose to limit the comparison to other assembly inference techniques to generative models.